# UniDiff: Spectral–Spatial Vision Models under Unified Diffusion

## Abstract

We present UniDiff, a general-purpose vision backbone driven by a unified diffusion operator. Under a single diffusion-semigroup framework, we construct a parallel Spatial–Spectral Diffusion module: the spectral path applies a heat-kernel multiplier to achieve global low-pass homogenization, while the spatial path performs anisotropic diffusion to preserve boundaries. Both paths are conditioned on a shared latent control field that predicts their parameters, and we impose PDE residual and energy-consistency constraints (with boundary consistency as auxiliary) at the operator level to pin the two steps to the same physical clock, eliminating the smear–sharpen counteraction and time-scale drift. Compared with self-attention, UniDiff attains efficient and interpretable global modeling. On ImageNet-1K Russakovsky et al. (2015), UniDiff achieves 84.2/84.8/85% Top-1 accuracy at the Tiny/Small/Base scales. On COCO Lin et al. (2014) dataset object detection/instance segmentation and ADE20K Zhou et al. (2017) dataset semantic segmentation, it also exhibits advantages in parameter number, FLOPs, and inference throughput over peer baselines. With computational complexity $\mathcal{O}(N \log N)$ and linear memory, UniDiff provides a unified and controllable spectral–spatial modeling paradigm, delivering robust representations for classification, detection, and segmentation.

## 1 Introduction

Visual representation models have served as the foundation of modern computer vision, driving progress across a wide range of tasks, including image classification, object detection, and semantic segmentationLiu et al. (2021); Wang et al. (2025b); Liu et al. (2022). Their effectiveness directly determines the performance and robustness of downstream applications such as autonomous drivingGeiger et al. (2012) and medical image analysisLitjens et al. (2017).

Existing vision backbones face two fundamental limitations. First, the computational burden of self-attention leads to heavy cost at scale; second, when edge artifacts and cross-block leakage arise, there is a lack of interpretable and controllable geometric constraints. In addition, current backbones generally lack an explicit modeling of frequency response—most improvements remain at the level of discrete engineering tricks and lack theoretical guarantees consistent with continuous-time evolution; consequently, they cannot explain the equivalence/commutativity of multi-module compositions along the time dimension. Furthermore, to obtain both globally consistent semantics and boundary-level geometric fidelity, a common practice is to split them into "global modules and boundary modules" in parallel or series. However, without a unified generative mechanism and physics-consistent coupling, training often suffers from smear–sharpen counteraction, indeterminate time scales/intensities, and boundary leakage, making it difficult to yield representations that are stable and scalable to downstream tasks. We propose a diffusion-driven vision architecture that unifies feature propagation under a single diffusion semigroup. In the spectral domain, a heat-kernel multiplier implements global low-pass filtering; in the spatial domain, anisotropic diffusion propagates tangentially along boundaries while suppressing normal directions. Both paths are parameterized by a shared latent control field that generates their respective parameters. At the operator level, we further impose a PDE residual and energy-consistency to ensure the two steps are consistent approximations of the same generative operator, and we regularize boundary behavior via zero normal flux (Neumann) conditions. Each layer costs $\mathcal{O}(N \log N)$ for the spectral path plus $\mathcal{O}(N)$ for the spatial path, which is markedly more efficient than global self-attention at $\mathcal{O}(N^2)$. This design, using

Figure 1: fOverall architecture of our UniDiff with an encoder-decoder structure.

few parameters and fully differentiable operators, improves representation quality and downstream performance through physics-informed constraints while maintaining high computational efficiency.

In summary, our contributions are as follows: 1. We propose a unified diffusion vision backbone that leverages physics-guided diffusion priors to enhance model interpretability under low computational complexity $\mathcal{O}(N \log N)$. 2. Grounded in physical diffusion theory, we unify spectral global homogenization and spatial boundary fidelity within a single diffusion semigroup framework. 3. We introduce a shared latent control field with a dual-head parameterization that predicts the spectral gate $k(\omega)$ and the spatial diffusivity tensor $D(x)$; at the operator level, we impose PDE residual and energy-consistency constraints. 4. We design the Spatial–Spectral Diffusion Block , which parallel-couples a spatial diffusion operator with a spectral diffusion operator. Our framework is shown in Fig. 1. The proposed S2D Block is detailed in Fig. 2.

## 2 RELATED WORK

### 2.1 CNNs AND VISION TRANSFORMERS

Convolutional neural networks (CNNs) have long served as the core foundation of visual recognition. Since ResNet He et al. (2015) established the paradigm of deep networks, large-kernel variants such as ParC-Net Zhang et al. (2022) have further strengthened global context modeling. The introduction of Vision Transformers (ViTs) Dosovitskiy et al. (2021) brought global self-attention to visual modeling, with hierarchical designs like Swin Transformer Liu et al. (2021) improving scalability; lightweight models such as MobileViT Mehta & Rastegari (2022) and EfficientFormer Li et al. (2022b) strike a favorable balance between accuracy and efficiency. Recently, state space models (SSMs) have offered a new direction for visual representation: VMamba Liu et al. (2024) introduces the Mamba architecture into vision backbones to unify long-range dependencies and local features; MambaVision Hatamizadeh & Kautz (2025) builds an end-to-end, purely SSM-driven vision architecture that substantially enhances global modeling; and GroupMamba Shaker et al. (2025) improves feature interaction and representational efficiency via grouped state space design.

### 2.2 BIOLOGY- AND PHYSICS-INSPIRED VISION MODELS

A complementary line of research seeks inspiration from biology and physics. Spiking Neural Networks (SNNs) Lee et al. (2016) simulate neuronal firing to process event-based vision, while fovea-inspired designs such as the Focal Transformer Yang et al. (2021) and TransNeXt ? mimic the non-uniform acuity of the human retina. From the physics perspective, anisotropic diffusion filteringWeickert et al. (1998) and diffusion probabilistic models Ho et al. (2020) highlight the role of partial differential equations and thermodynamic priors in feature propagation. QB-Heat Chen et al. (2022) further employs the heat equation to supervise masked image modeling, and vHeat Wang et al. (2025b) reformulates global context learning as a heat conduction operator, achieving efficient global coupling. While these works demonstrate the promise of biologically and physically grounded priors, they are either limited to low-level tasks or tailored to specific designs, leaving open the challenge of building a unified backbone that simultaneously ensures global semantic consistency and boundary fidelity under low computational complexity.

## 3 UNIFIED DIFFUSION VIEW

Many vision tasks face a dual challenge in representation: insufficient global consistency (fragmentation/semantic drift in large, low-texture regions) and geometric distortion at boundaries (oversmoothing and cross-boundary leakage). An ideal feature field should be low-frequency smooth within object interiors and piecewise discontinuous at object/occlusion boundaries. To this end, we co-model two formulations of the same diffusion process: in the spectral domain, a heat kernel provides stable low-pass filtering to reinforce global consistency; in the spatial domain, anisotropic

diffusion enables tangential propagation with normal-direction suppression to preserve boundary fidelity. Built on this, we develop a diffusion-unified vision framework: a spectral–spatial collaborative diffusion operator reshapes features globally and aligns them at boundaries under a common physical semantics, delivering robust representations for classification, detection, and segmentation while maintaining computational efficiency and interpretability.

**Notation.** Let the image lattice be $\Omega = \{1, \ldots, H\} \times \{1, \ldots, W\}$ and let $u : \Omega \to \mathbb{R}^C$ denote an image representation. We adopt anisotropic diffusion as the mother equation:

$$\frac{\partial u(\mathbf{x}, t)}{\partial t} = \nabla \cdot \big(D(\mathbf{x})\, \nabla u(\mathbf{x}, t)\big) \; \triangleq \; L\, u(\mathbf{x}, t), \qquad u(\cdot, t) = e^{\,tL} u(\cdot, 0), \tag{1}$$

where $D(\mathbf{x}) \in \mathbb{R}^{2 \times 2}$ is symmetric positive definite (SPD). We use Neumann boundary conditions (implemented as reflection padding). Equation 1 generates a dissipative semigroup with $\|e^{tL}\|_{2 \to 2} \leq 1$ for all $t \geq 0$.

**Energy and stability.** Define the discrete diffusion energy and by the discrete Green's identity A.5,

$$\mathcal{E}(u) = \tfrac{1}{2} \sum_{\mathbf{x} \in \Omega} \big(\nabla u(\mathbf{x})\big)^\top D(\mathbf{x})\, \nabla u(\mathbf{x}) \; \geq 0. \tag{2}$$

$$\langle u,\, \nabla \cdot (D \nabla u) \rangle = -\langle \nabla u,\, D \nabla u \rangle \leq 0 \; \Rightarrow \; \frac{d}{dt} \mathcal{E}(u(t)) \leq 0, \tag{3}$$

hence diffusion is energy non-increasing and $L_2$-nonexpansive whenever $D \succeq 0$, providing a stability guarantee for learnable approximations.

### 3.1 SPECTRAL FORMULATION: A ONE-STEP GLOBAL CONSISTENCY UPDATE

Under Neumann boundary conditions, the discrete Laplacian $\Delta$ is diagonalized by the 2D DCT. Let $\mathcal{F}/\mathcal{F}^{-1}$ be DCT/IDCT and $\Lambda(\omega)$ the Laplacian spectrum ( frequency index $\omega = (p, q)$). Approximating $D(\mathbf{x})$ locally by an average $\overline{D}$ (or over short time), the spectral solution of Eq. 1:

$$\widehat{u}(\omega, t) = \exp\Big(-\, t\, \omega^\top \overline{D}\, \omega\Big)\, \widehat{u}_0(\omega),$$

i.e., a heat-kernel low-pass. We replace the fixed quadratic form by a learnable nonnegative spectral attenuation $k(\omega) \geq 0$ and perform a near-closed one-step propagation:

$$\tilde{u} = \mathcal{F}^{-1}\Big[\underbrace{\exp\big(-\, t_g\, \Lambda(\omega)\, k(\omega)\big)}_{M(\omega) \in (0, 1]} \odot \mathcal{F}(u)\Big], \tag{4}$$

with spectral step $t_g > 0$ and element-wise product $\odot$. Since $|M(\omega)| \leq 1$, Eq. 4 is a contraction, yielding a single-pass, stable global low-frequency alignment. Minimal, physics-consistent parameterization. We use a radial + second-order angular basis:

$$k(\omega) = \sum_{i=1}^{M} \alpha_i\, \varphi_i(\|\omega\|) \;+\; \beta_1 \cos\big(2 \angle \omega\big) \;+\; \beta_2 \sin\big(2 \angle \omega\big), \quad \alpha_i, \beta. \geq 0, \tag{5}$$

where $\{\varphi_i\}$ partition frequency bands; $\cos 2\theta, \sin 2\theta$ are the lowest nontrivial angular harmonics (with $\pi$-periodicity) of 2D quadratic forms, capturing directional low-pass behavior aligned with scene structure. The spectral step suppresses low-frequency errors and aligns global context in one pass, mitigating scale drift and holes over large/low-texture regions. The directional terms dampen spurious high frequencies in off-structure directions, aiding long-range consistency.

### 3.2 SPATIAL FORMULATION: AN EDGE-PRESERVING EXPLICIT STEP

To preserve discontinuities, we adopt a boundary-aligned anisotropic tensor:

$$D(\mathbf{x}) = R\big(\theta(\mathbf{x})\big) \operatorname{diag}\big(\kappa_\|(\mathbf{x}),\, \kappa_\perp(\mathbf{x})\big) R\big(\theta(\mathbf{x})\big)^\top, \quad \kappa_\perp \leq \kappa_\|,\, \kappa. \geq 0, \tag{6}$$

where $\theta$ aligns with boundary tangents; reducing $\kappa_\perp$ along boundary normals suppresses cross-boundary diffusion. We then take a small explicit Euler step: $u^+ = \tilde{u} + \Delta t\, \nabla \cdot \big(D(\mathbf{x})\, \nabla \tilde{u}\big)$, with $\Delta t$ chosen under a conservative CFL bound (e.g., $\Delta t \leq [4 \max_{\mathbf{x}} \kappa_\|(\mathbf{x})]^{-1}$). By Eq. 3, $\mathcal{E}(u^+) \leq \mathcal{E}(\tilde{u})$ and $\|u^+\|_2 \leq \|\tilde{u}\|_2$: the spatial step is also energy non-increasing and non-expansive. The explicit step, with tangent-permissive, normal-suppressive diffusion, preserves geometric discontinuities and thin structures, alleviating edge-fattening and cross-instance leakage.

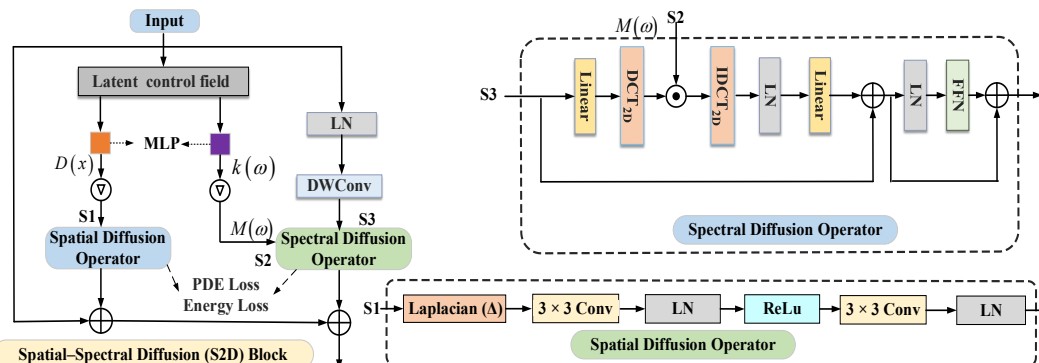

Figure 2: Overview of Spatial–Spectral Diffusion Block (S2D Block). The block comprises two parallel paths—a spatial diffusion operator and a spectral diffusion operator—driven by a latent control field. The control field is fed into two disjoint MLP heads that respectively predict a spatial diffusion tensor $D(x)$ and a spectral attenuation $k(\omega)$. **Spatial Diffusion Operator** applies anisotropic diffusion by evaluating the discrete operator $\nabla \cdot (D(x)\nabla u)$, i.e., the anisotropic form of the Laplacian $\Delta$. **Spectral Diffusion Operator** diagonalizes the Laplacian in the DCT domain and performs a heat-kernel update parameterized by $k(\omega)$ to obtain $M(\omega)$, followed by $\text{IDCT}_{2D}$ back to the spatial domain. Together, the two paths constitute a learnable, gated unified diffusion operator.

### 3.3 A UNIFIED DIFFUSION OPERATOR IN SPECTRAL AND SPATIAL DOMAINS

To ensure methodological coherence and reproducibility, we first establish that the one–step spectral update and the explicit spatial step are two approximations of the same generator $e^{tL}$, rather than a heuristic concatenation. Without this common origin, the energy dissipation and $L_2$-non-expansive properties of each step would not be inherited by their composite mapping, thereby risking over-smoothing or artifact amplification during training. Moreover, the learnable spectral kernel $k(\omega)$ and the spatial anisotropic diffusion tensor $D(\mathbf{x})$ would lack a unifying physical bridge, leading to parameter non-identifiability and intrinsic conflicts. For a detailed proof, see Appendix A.4.

### 3.4 LATENT CONTROL FIELD

We define $g(x)$ as a per-pixel diffusion driving map (latent control field). It provides data-driven evidence and constraints for "where/when to enable or suppress the diffusion operator, along which direction to propagate, and at what strength to evolve." $g(x)$ can simultaneously predict the "spectral attenuation head in the frequency domain" and the "diffusion tensor head in the spatial domain." The diffusion equation is equivalent, in the frequency domain, to exponential decay of DCT modes (heat-kernel low-pass), and, in the spatial domain, to local propagation measured by the anisotropic tensor $D(x)$. Therefore, the evidence required to determine "which frequencies to suppress" and "along which geometric direction to diffuse is essentially the same class of local cues: geometric structure and uncertainty. $g(x)$ is exactly a unified encoding of these two kinds of cues, and thus the same $g(x)$ can consistently control both heads: **Frequency side:** the scene's overall scale, texture richness, and dominant-direction statistics jointly determine the relative weights of low/high-frequency suppression and the strength of directional suppression, thereby fixing the radial and angular coefficients of the spectral attenuation $k(\omega)$; **Spatial side:** the boundary direction and the anisotropy strength determine the concrete ratio of "tangential pass and normal suppression," thereby fixing the principal-axis direction and principal values of the diffusion tensor $D(x)$.

The angular parameters in the frequency domain can be aligned with the principal axes of $D(x)$, and the spectral step size and the spatial step size can be regarded as an allocation of the same diffusion time budget; therefore, using a single $g(x)$ to predict $k(\omega)$ and $D(x)$ is theoretically self-consistent and identifiable, and can collaboratively achieve global consistency and boundary fidelity. In summary, the evidence for "which frequencies to suppress and along which directions to diffuse" is uniformly encoded in $g(x)$. Accordingly, we map the same latent control field $g(x)$ to two learnable heads: the spectral attenuation $k(\omega)$ and the spatial diffusivity tensor $D(x)$. Operating under a single diffusion semigroup with a unified time step $t_g$, the two heads act cooperatively via a spectral–

spatial splitting: the spectral path enforces global homogenization, while the spatial path preserves boundary fidelity. We next detail their identifiable parameterizations and discrete implementations.

**1. Generating the spectral attenuation head $k(\omega)$ from $g(x)$.** We adopt a minimal yet sufficient "radial + second-order angular" basis to ensure low-pass behavior and directional expressiveness:

$$k(\omega) = \sum_{i=1}^{M} \alpha_i \, \varphi_i(\|\omega\|) \; + \; \beta_1 \cos\big(2\angle\omega\big) \; + \; \beta_2 \sin\big(2\angle\omega\big), \quad \alpha_i, \beta_1, \beta_2 \geq 0. \tag{7}$$

Here $\{\varphi_i\}_{i=1}^{M}$ are radial basis functions (uniform or logarithmic bands). The $\cos 2\theta$ and $\sin 2\theta$ terms are the lowest nontrivial angular harmonics ($\pi$-periodic) of 2D quadratic forms, capturing directional low-pass effects. Nonnegativity guarantees the spectral multiplier

$$M(\omega) = \exp\big[ - \, t_g \, \Lambda(\omega) \, k(\omega)\big] \in (0, 1], \tag{8}$$

thus defining a dissipative operator. $\Lambda(\omega)$ denotes the DCT eigen-spectrum of the discrete Laplacian, and $t_g > 0$ is the spectral time step. **From $g$ to parameters.** We apply global average pooling to obtain $\bar{g}$, and use a small multilayer perceptron to predict $\mathrm{MLP}_k(\bar{g}) = \{\alpha_1, \ldots, \alpha_M, \beta_1, \beta_2, t_g\}$, followed by a single spectral propagation

$$\tilde{u} = \mathcal{F}^{-1}\big(M(\omega) \odot \mathcal{F}(u)\big), \tag{9}$$

which yields a one-pass, stable low-pass with directional consistency for global context aggregation.

**2. Generating the spatial diffusion tensor head $D(x)$ from $g(x)$.** We use the symmetric positive definite (SPD) "rotation + diagonal" minimal form for interpretability and numerical stability:

$$D(x) = R\big(\theta(x)\big) \mathrm{diag}\big(\kappa_\parallel(x), \kappa_\perp(x)\big) R\big(\theta(x)\big)^\top, \quad \kappa_\perp \geq 0, \;\; \kappa_\parallel \geq \kappa_\perp. \tag{10}$$

Here $\theta(x)$ approximates the boundary tangent direction; $\kappa_\parallel$ and $\kappa_\perp$ control tangential and normal diffusion strengths, respectively. To robustly satisfy SPD and ordering, we use $\kappa_\perp = \mathrm{softplus}(a)$ and $\kappa_\parallel = \kappa_\perp + \mathrm{softplus}(b)$. **From $g$ to parameters.** A shallow convolutional head $\mathrm{Head}_D$ predicts, per pixel, $\mathrm{Head}_D\big(g(x)\big) = \{\theta(x), \kappa_\parallel(x), \kappa_\perp(x)\}$, from which we assemble $D(x)$ and perform a single explicit Euler step, ensuring (CFL condition) energy monotonicity and $L_2$-nonexpansiveness.

$$u^+ = \tilde{u} + \Delta t \, \nabla\cdot\big(D(x) \, \nabla\tilde{u}\big), \qquad \Delta t \leq \frac{1}{4 \, \max_x \kappa_\parallel(x) + \varepsilon}, \tag{11}$$

### 3.5 Spectral–Spatial Discretization under a Unified Diffusion Semigroup

To realize learnable, provable, and efficient representations under a single diffusion semigroup, we provide discrete implementations for a spectral attenuation head and a spatial anisotropic diffusion head, and adopt Neumann (reflective) boundary conditions to match natural image behavior.

**Spectral Attenuation Head.** We set the spectral multiplier $M(\omega)$ to be a heat kernel and perform a stable, per-frequency low-pass update on the input feature $u$:

$$\tilde{u} = \mathcal{F}^{-1}\big( \exp[-t_g \Lambda(\omega) k(\omega)] \odot \mathcal{F}(u)\big), \tag{12}$$

where $t_g > 0$ is the unified time step, $\Lambda(\omega)$ denotes the symbol (eigen-spectrum) of the discrete Laplacian, and $k(\omega) \geq 0$ is a learnable spectral gate. To obtain directional control while preserving dissipativity, we parameterize $k(\omega)$ with a radial + second-order angular basis and enforce $0 < M(\omega) \leq 1$. For the discrete implementation, we diagonalize the discrete Laplacian using DCT/IDCT, constructed in the DCT domain $M[p, q] = \exp\big[ - t_g \Lambda_{p,q} k_{p,q}\big]$, apply pointwise multiplication, and map back to the spatial domain via IDCT. This naturally corresponds to Neumann (reflective) boundaries. The time complexity is $O(N \log N)$.

**Spatial Diffusion Head.** The spatial branch employs the anisotropic diffusion operator $\nabla\cdot(D\nabla\cdot)$. At each pixel, we parameterize the diffusion tensor as

$$D = R(\theta) \mathrm{diag}(\kappa_\parallel, \kappa_\perp) R(\theta)^\top, \qquad \kappa_\parallel \geq \kappa_\perp \geq 0,$$

from which the effective conductivities along the horizontal/vertical axes $c_x, c_y$ are obtained. To ensure symmetry and dissipativity, the four face conductances (E/W/N/S) are built using the harmonic mean . With a flux-difference approximation of the divergence, an explicit Euler update reads

$$u^+ = \tilde{u} + \Delta t \, [\, w_E(\tilde{u}_{i+1,j} - \tilde{u}_{i,j}) - w_W(\tilde{u}_{i,j} - \tilde{u}_{i-1,j}) + w_N(\tilde{u}_{i,j+1} - \tilde{u}_{i,j}) - w_S(\tilde{u}_{i,j} - \tilde{u}_{i,j-1}) \,],$$

where $w_{\{\cdot\}}$ are the face conductances. To guarantee energy decay and $L_2$-nonexpansiveness, $\Delta t$ must satisfy the corresponding CFL upper bound. The spatial branch has time complexity $O(N)$.

Neumann boundaries are implemented via reflection padding; the $\varepsilon > 0$ term in the harmonic mean avoids divide-by-zero. The scheme consists only of per-pixel algebra and four-neighborhood differences, making it easy to integrate as a fully differentiable network layer. The spectral and spatial branches are coupled in parallel under the same time step $t_g$: the spectral path provides global homogenization and noise shaping, while the spatial path enforces boundary fidelity and anisotropic alignment. Together, they deliver efficient and interpretable modeling under a unified diffusion view. Detailed derivations are provided in the Appendix A.6.

### 3.6 Semigroup Consistency: Joint Spectral and Spatial Constraints

We denote the image feature by $u : \Omega \to \mathbb{R}$, where $|\Omega|$ is the number of pixels. The diffusion operator $Lu \triangleq \nabla \cdot \big( D(x) \nabla u \big), \qquad D(x) \succeq 0$, governs anisotropic propagation in the spatial domain. One spectral step is realized by a heat-kernel multiplier

$$\tilde{u}^{\ell+1} = \mathcal{F}^{-1}\Big( \exp[-t_g\, \Lambda(\omega)\, k(\omega)] \odot \mathcal{F}(u^\ell) \Big), \qquad k(\omega) \geq 0, \ t_g > 0, \tag{13}$$

followed by one explicit spatial step $u^{\ell+1} = \tilde{u}^{\ell+1} + \Delta t\, L\tilde{u}^{\ell+1}$. These two steps constitute a Lie–Trotter splitting approximation of the same diffusion semigroup $e^{tL}$. Without appropriate coupling constraints, the parameter triplet $\{t_g, k(\omega), D(x)\}$ may exhibit identifiability issues and time-scale drift: different $(k, D)$ combinations can produce similar appearances yet correspond to physically inconsistent propagation, often manifesting in training as a cancellation between global smoothing and local diffusion, boundary fattening, and cross-object leakage. To bind the two heads to the same generator and to impose the correct boundary behavior, we minimize three complementary losses: a PDE residual (strong form), an energy consistency (weak form), and a boundary flux penalty.

#### 3.6.1 Partial differential equation (PDE) residual consistency (strong form).

By first-order time discretization (Taylor expansion), $e^{t_g L} u = u + t_g\, Lu + \mathcal{O}(t_g^2)$. Since our spectral step $\tilde{u}$ approximates $e^{t_g L_g} u$, the begin-of-step quantities should satisfy $(\tilde{u}^{\ell+1} - u^\ell)/t_g \approx L\, u^\ell$. We thus define a pixel-wise residual and take its unweighted average:

$$\mathcal{L}_{\text{PDE}} = \frac{1}{|\Omega|} \sum_{x \in \Omega} \left| \frac{\tilde{u}^{\ell+1}(x) - u^\ell(x)}{t_g} - [L\, u^\ell](x) \right|. \tag{14}$$

This term aligns the "time derivative induced by the spectral step" with the "spatial generator" at the same time instant. Consequently, the spectral time budget $t_g k(\cdot)$ and the diffusion tensor $D(x)$ are calibrated to share a consistent physical meaning, preventing them from compensating each other by free rescaling. It also suppresses the accumulation of splitting errors (from $[L_g, L_\ell]$) to first order, stabilizing the composite dynamics. Empirically, $\mathcal{L}_{\text{PDE}}$ reduces the counteracting pattern of "over-smoothing globally vs. over-stretching locally," yielding fewer ringing/halo artifacts and improved long-range scale consistency.

#### 3.6.2 Energy consistency (weak form).

For any field $u$, the diffusion energy admits two equivalent expressions: the spectral-side quadratic form, where each frequency is attenuated by $\Lambda(\omega)k(\omega)$, and the spatial-side anisotropic Dirichlet energy $\langle \nabla u, \bar{D} \nabla u \rangle$ with $\bar{D} = \frac{1}{|\Omega|} \sum_x D(x)$. Let $\hat{u}^\ell = \mathcal{F}(u^\ell)$. Define as:

$$\mathcal{E}_{\text{spec}}(u^\ell) = \sum_\omega \Lambda(\omega)\, k(\omega)\, |\hat{u}^\ell(\omega)|^2, \qquad \mathcal{E}_{\text{space}}(u^\ell) = \sum_\omega (\omega^\top \bar{D} \omega)\, |\hat{u}^\ell(\omega)|^2, \tag{15}$$

and the weak-form alignment loss $\mathcal{L}_{\text{energy}} = \left| \mathcal{E}_{\text{spec}}(u^\ell) - \mathcal{E}_{\text{space}}(u^\ell) \right|$.

Built upon Plancherel duality, this term globally matches the spectral time budget $\Lambda k$ with the spatially averaged anisotropy $\bar{D}$, thereby eliminating overall scale drift so that $t_g k(\cdot)$ and $\bar{D}$ share a unified time unit. When $k(\omega)$ contains second-order angular components, $\mathcal{L}_{\text{energy}}$ further aligns the principal directions of the spectral kernel with those of $\bar{D}$, reducing direction mismatches that cause structural artifacts. Compared with the point-wise residual, this weak form is more robust to noise/occlusions and provides a stable global pull in early training, complementing $\mathcal{L}_{\text{PDE}}$.

#### 3.6.3 Boundary flux constraint (zero normal flux).

Anisotropic diffusion imposes a zero normal flux boundary condition $n^\top D\nabla u = 0$. Rather than penalizing total variation over the entire image, we only suppress the normal component of the flux on detected boundary pixels while allowing tangential propagation to maintain within-contour coherence. Let $\mathcal{E} \subset \Omega$ be the boundary set (obtained, e.g., by a structure-tensor or gradient-magnitude threshold) with unit normal $n(p)$. We impose the soft constraint

$$\mathcal{L}_{\text{edge}} = \frac{1}{|\mathcal{E}|} \sum_{p \in \mathcal{E}} \big| n(p)^\top D(p) \nabla u^\ell(p) \big|. \tag{16}$$

This is the discrete Neumann condition for our anisotropic diffusion: it suppresses cross-object normal flow while preserving tangential spreading, thereby avoiding edge fattening and leakage without harming contour-consistent smoothing. In boundary regions, $\mathcal{L}_{\text{PDE}}$ enforces generator agreement whereas $\mathcal{L}_{\text{edge}}$ specifies the admissible direction of propagation; the two are complementary. These terms bind the spectral and spatial heads to the same diffusion generator. They jointly calibrate the time budget and anisotropy, reduce splitting error and parameter ambiguity, and achieve long-range scale consistency without sacrificing boundary sharpness.

# 4 EXPERIMENT

## 4.1 EXPERIMENTAL RESULTS:

**Network Architecture.** We build a family of UniDiff models, including UniDiff-Tiny, UniDiff-Small, and UniDiff-Base. An overview of the UniDiff architecture is shown in Fig. 1, S2D block is shown in Fig. 2 and detailed configurations are provided in Appendix A.8.

**Image Classification** On ImageNet-1K Russakovsky et al. (2015), we first compare parameter count, FLOPs, test throughput, and accuracy in Table 1. For the Tiny scale, under comparable parameters and FLOPs, our UniDiff-T attains a Top-1 accuracy of **83.8%**; this advantage becomes more pronounced at the Small and Base scales. Specifically, the Small model reaches **84.5%**, surpassing SOTA results; at the Base scale, despite markedly reduced parameters and FLOPs, we still achieve **84.7%**. In terms of computational efficiency, UniDiff maintains state-of-the-art accuracy while keeping FLOPs and parameter counts low, yielding a superior accuracy–efficiency trade-off. This is particularly evident at the Small and Base scales: Although MambaVision Hatamizadeh & Kautz (2025) and InceptionNeXt Yu et al. (2024b) achieve higher FPS, they do so at the cost of lower accuracy and are therefore not comparable at the same accuracy level. TransNeXt attains accuracy close to ours, but with substantially larger parameter numbers and FLOPs, and without an advantage in throughput. Overall, our model delivers higher inference efficiency with a smaller compute and memory budget, while matching or exceeding the accuracy of competing methods. More comprehensive comparisons with additional SOTA methods are provided in Table 5 of Appendix A.9.

**Downstream Task Evaluation:** On COCO dataset Lin et al. (2014), we evaluate ImageNet-1KRussakovsky et al. (2015) pre-trained ours model with a Mask R-CNN He et al. (2017) head under a $1\times$ training schedule (Table 4). All three scales achieve the best $\text{AP}^b/\text{AP}^m$ while maintaining high FPS: **Tiny** 50.2/45.9, **Small** 51.3/45.8, and **Base** 51.5/46.1. On ADE20K Zhou et al. (2017), using UperNet Xiao et al. (2018) for semantic segmentation, all three models also obtain the best mIoU: **Tiny** 51.4, **Small** 52.4, and **Base** 53.2. Across these tasks, our FPS is only slightly lower than vHeat Wang et al. (2025b), yet the accuracy gains are more substantial, indicating that downstream performance can be achieved without sacrificing efficiency. More comprehensive comparisons with additional SOTA methods are provided in Table 6 and Table 7 of Appendix A.9.

## 4.2 FIXED VS. LEARNED DIFFUSION PARAMETERS

We conduct ablations of our Unidiff-tiny model on ImageNet-1K dataset Russakovsky et al. (2015), comparing the spectral gate $k(\omega)$ and the spatial diffusivity tensor $D(x)$ under two settings: fixed versus learned. In the fixed-parameter setting, we sweep a range of values (see Appendix A.10, Fig. 3 and Fig. 4): the Top-1 accuracy lies in $81.2\% - 83.0\%$ when fixing $k(\omega)$, and in $80.5\% - 82.5\%$ when fixing $D(x)$. When both are learned, the Top-1 rises to $83.8\%$. We attribute this gain to their physically interpretable adaptivity. First, $D(x)$ is the key coefficient in the anisotropic diffusion operator $\nabla \cdot (D(x)\nabla u)$ and can be interpreted as a local effective diffusivity/thermal conductivity. A fixed $D(x)$ cannot simultaneously balance in-region smoothing and across-edge inhibition, leading to sample-dependent optima; a learned $D(x)$ reduces diffusion near edges while increasing it in homogeneous areas, achieving a dynamic trade-off between edge fidelity and noise suppression. Second, the spectral heat-kernel multiplier (Eq. 8) implements a frequency-dependent rescaling of effective diffusion time/strength. Because image spectra and noise characteristics are markedly non-stationary across samples and classes, a fixed $k(\omega)$ cannot adapt to both "high-frequency–rich/noisy" and "low-frequency–dominant/ sparse-texture" regimes. In contrast, a learned $k(\omega)$ performs frequency-wise adaptive shaping conditioned on semantic features, preserving discriminative high frequencies while attenuating spurious high-frequency noise, thereby yielding additional gains.

| Method | Image size | #Param. | FLOPs | Test Throughput (img/s) | ImageNet top-1 acc. (%) |
|---|---|---|---|---|---|
| LSNet Wang et al. (2025a) | $224^2$ | 23 M | 1.3 G | 3996 | 80.3 |
| Swin-T Liu et al. (2021) | $224^2$ | 28 M | 4.6 G | 1242 | 81.3 |
| ConvNeXt-T Liu et al. (2022) | $224^2$ | 29 M | 4.5 G | 1198 | 82.1 |
| DCFormer-SW-T Li et al. (2023) | $512^2$ | 28 M | 4.5 G | – | 82.1 |
| Vim-S Zhu et al. (2024) | $224^2$ | 26 M | 5.3 G | 811 | 81.4 |
| vHeat-TWang et al. (2025b) | $224^2$ | 29 M | 4.6 G | 1514 | 82.2 |
| InceptionNeXt-T Yu et al. (2024b) | $224^2$ | 28 M | 4.2 G | 2900 | 82.3 |
| VMamba-T Liu et al. (2024) | $224^2$ | 30 M | 4.9 G | 1282 | 82.6 |
| MambaVision-T Hatamizadeh & Kautz (2025) | $224^2$ | 35 M | 5.1 G | **5990** | 82.7 |
| GroupMamba Shaker et al. (2025) | $224^2$ | 23 M | 4.5 G | – | 83.3 |
| BiFormer-S Zhu et al. (2023) | $224^2$ | 26 M | 4.5 G | – | 83.8 |
| TransNeXt-T Shi (2024) | $224^2$ | 28 M | 5.7 G | 756 | 84.0 |
| Ours | $224^2$ | 30 M | 4.7 G | 1472 | **84.2** |
| Swin-S Liu et al. (2021) | $224^2$ | 50 M | 8.7 G | 720 | 83.0 |
| ConvNeXt-S Liu et al. (2022) | $224^2$ | 50 M | 8.7 G | 687 | 83.1 |
| DCFormer-SW-S Li et al. (2023) | $512^2$ | 50 M | 8.7 G | – | 82.9 |
| InceptionNeXt-S Yu et al. (2024b) | $224^2$ | 49 M | 8.4 G | 1750 | 83.5 |
| vHeat-S Wang et al. (2025b) | $224^2$ | 50 M | 8.5 G | 945 | 83.6 |
| VMamba-S Liu et al. (2024) | $224^2$ | 53 M | 8.7 G | 843 | 83.6 |
| MambaVision-S Hatamizadeh & Kautz (2025) | $224^2$ | 50 M | 7.5 G | **4700** | 83.3 |
| GroupMamba Shaker et al. (2025) | $224^2$ | 45 M | 7.0 G | – | 83.9 |
| BiFormer-B Zhu et al. (2023) | $224^2$ | 57 M | 9.8 G | 394 | 84.3 |
| TransNeXt-S Shi (2024) | $224^2$ | 50 M | 10.3 G | – | 84.7 |
| Ours | $224^2$ | 53 M | 8.5 G | 921 | **84.8** |
| Swin-B Liu et al. (2021) | $224^2$ | 88 M | 15.4 G | 456 | 83.5 |
| ConvNeXt-B Liu et al. (2022) | $224^2$ | 89 M | 15.4 G | 439 | 83.8 |
| DCFormer-SW-B Li et al. (2023) | $512^2$ | 88 M | 15.4 G | – | 83.5 |
| Vim-B Zhu et al. (2024) | $224^2$ | 98 M | 19.0 G | 294 | 83.2 |
| vHeat-B Wang et al. (2025b) | $224^2$ | 69 M | 11.2 G | 661 | 84.0 |
| InceptionNeXt-B Yu et al. (2024b) | $224^2$ | 87 M | 14.9 G | 1244 | 84.0 |
| VMamba-T Liu et al. (2024) | $224^2$ | 89 M | 15.4 G | 645 | 83.9 |
| MambaVision-T Hatamizadeh & Kautz (2025) | $224^2$ | 97 M | 15.0 G | **3670** | 84.2 |
| GroupMamba Shaker et al. (2025) | $224^2$ | 57 M | 14.0 G | – | 84.5 |
| TransNeXt-BShi (2024) | $224^2$ | 90 M | 18.4 G | 297 | 84.8 |
| Ours | $224^2$ | 72 M | 11.4 G | 635 | **85.0** |

Table 1: Comparison with state-of-the-art methods on ImageNet-1K Russakovsky et al. (2015).

### 4.3 Spatial–Spectral Diffusion Operator Analysis

**Effective Receptive Field of the S2D Operator.** As shown in the Appendix Fig. 7, we compare the ERF of Ours (S2D) with other methods across four stages. Unlike the baselines, S2D expands from Stage 2 into a continuous, smooth, near-circular global field while retaining mild directional anisotropy near boundaries to better respect structure; it simultaneously avoids blocky and lobed artifacts and mitigates over-smoothing. Overall, under comparable or lower computational cost, S2D achieves an ERF that expands rapidly, remains well-shaped, and decays smoothly—reflecting a more physics-consistent information perception and a more natural visual representation, thereby validating the effectiveness of our design.

**Computational Complexity of the S2D Operator.** Our design substantially reduces computational and memory costs without sacrificing global modeling capacity. Specifically, the spectral diffusion path leverages the diagonalization property of DCT/FFT to convert the update into element-wise multiplications in the frequency domain, yielding a per-layer complexity of $\mathcal{O}(N \log N)$. The spatial diffusion path approximates $\nabla \cdot (D \nabla \cdot)$ using sparse local stencils / depthwise separable convolutions, with complexity $\mathcal{O}(N)$. When the two paths are parallel-coupled, the dominant cost of S2D is $\mathcal{O}(N \log N) + \mathcal{O}(N)$, which is markedly lower than the $\mathcal{O}(N^2)$ cost required by standard global self-attention. Meanwhile, S2D requires only linear intermediate storage and does not rely on large attention matrices, leading to higher throughput and lower compute/memory overhead. A detailed derivation of the computational complexity is provided in Appendix A.7.

**Representation Analysis of the Spatial–Spectral Diffusion Operator.** To assess the proposed operator's ability to extract discriminative features, we compare four settings on diverse natural images in Appendix Fig. 6: Self-Attention (baseline), Spatial Only (spatial diffusion operator only), Spectral Only (spectral diffusion operator only), and Spatial–Spectral (S2D), which combines both. Detailed analysis is provided in Appendix A.12.1. Overall, the experiments indicate that the spatial path provides boundary alignment and anisotropic structural constraints, whereas the spectral

| Spectral Operator | Spatial Operator | PDE Loss | Energy Loss | Boundary Loss | top-1 acc | $AP^b$ | $AP^m$ | mIoU |
|---|---|---|---|---|---|---|---|---|
| – | – | – | – | – | 79.75 | 42.4 | 38.7 | 43.9 |
| ✓ | – | – | – | – | 82.35 | 45.2 | 41.3 | 46.9 |
| ✓ | ✓ | – | – | – | 83.45 | 47.9 | 43.2 | 49.1 |
| ✓ | ✓ | ✓ | – | – | 83.82 | 48.9 | 44.8 | 50.4 |
| ✓ | ✓ | ✓ | ✓ | – | 84.13 | 50.1 | 45.7 | 51.3 |
| ✓ | ✓ | ✓ | ✓ | ✓ | 84.20 | 50.2 | 45.9 | 51.4 |

Table 2: Ablation results for the Tiny model on ImageNet-1K Russakovsky et al. (2015) classification (Top-1), COCO Lin et al. (2014) object detection and instance segmentation ($AP_b/AP_m$), and ADE20K Zhou et al. (2017) semantic segmentation (mIoU).

path delivers global homogenization and noise shaping. Their synergy within a unified diffusion framework significantly improves both the representational power and interpretability of the model.

## 4.4 ABLATION STUDY

Our ablation results are summarized in Table 2. Starting from a self-attention baseline, we progressively add our operators and loss. We evaluate four metrics across three datasets: Top-1 on ImageNet-1K Russakovsky et al. (2015), $AP_b/AP_m$ on COCO Lin et al. (2014) object detection and instance segmentation, and mIoU on ADE20K Zhou et al. (2017) semantic segmentation. **Spectral diffusion operator — an efficient global homogenizer.** (See Table 2, row 1, 2.) By diagonalizing the Laplacian via DCT/FFT, the spectral path converts the update into a per-frequency heat-kernel modulation, establishing long-range dependencies at a cost of $\mathcal{O}(N \log N)$. This efficiently couples global information and suppresses spurious high-frequency noise, providing more coherent global semantics for classification and downstream tasks. When used alone, however, it lacks boundary awareness. **Spatial diffusion operator — boundary fidelity and anisotropy.** (See Table 2, row 2, 3.) Adding the spatial path markedly improves detection and segmentation: anisotropic diffusion propagates along structural principal directions while suppressing cross-edge leakage, aligning discriminative responses with object contours and yielding more focused attention regions for classification. **PDE Residual Loss — synchronizing the two paths under the same PDE/time step.** (See Table 2, rows 3, 4.) Using the two operators jointly further boosts all metrics. When the spectral step (controlled by $k(\omega)$) and the spatial step (controlled by $D(x)$) advance at different paces, the PDE residual $\mathcal{L}_{\text{PDE}}$ magnifies their output mismatch and forces a coordinated adjustment that aligns their implicit time scales to the same physical step $t_g$. Likewise, if one path tends to over-smooth while the other tends to sharpen, the requirement to produce a single-step update consistent with the same PDE and the same $t_g$ penalizes such "smear–pull" counteraction and aligns the magnitude and direction of the two updates. In essence, $\mathcal{L}_{\text{PDE}}$ imposes a physics-consistent coupling so that the two operators no longer act independently but jointly approximate the same diffusion time propagator. **Energy Consistency Loss — enforcing monotone dissipation.** (See Table 2, rows 4, 5.) This loss couples the update magnitudes of the two operators under the same dissipation law, suppressing oscillation and over-sharpening while avoiding over-smoothing. Compared with using only the PDE residual, energy consistency further tightens the scale and direction of the dual updates, and it is lightweight with negligible extra compute. **Boundary Consistency Loss (Neumann/reflective) — steady gains near boundaries.** (See rows 5, 6.) Although the incremental gain is modest, enforcing boundary consistency improves boundary quality in dense prediction and reduces sporadic artifacts.

## 5 CONCLUSION

We present UniDiff, a diffusion-driven vision backbone that systematically injects physical priors into visual modeling while balancing efficiency and interpretability. Its core module, **S2D**, parallel-couples a spectral diffusion operator with a spatial diffusion operator. Theoretically, within a unified diffusion-semigroup framework, we provide a provable realization via Lie–Trotter/Strang splitting and impose PDE-residual and energy-consistency constraints at the operator level. Computationally, each layer costs $\mathcal{O}(N \log N)$ in the spectral path plus $\mathcal{O}(N)$ in the spatial path, yielding greater efficiency than self-attention $\mathcal{O}(N^2)$. On ImageNet-1K Russakovsky et al. (2015), UniDiff attains **83.8%/84.5%/84.7%** Top-1 accuracy at the Tiny/Small/Base scales, while delivering superior inference throughput, FLOPs, and parameter counts compared with common baselines at matched accuracy. On downstream detection and segmentation, UniDiff matches or surpasses larger backbones under standard training schedules, demonstrating strong transferability.

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

# A APPENDIX

## A.1 LLM USAGE STATEMENT

Large language models (e.g., OpenAI ChatGPT) were used only for English writing assistance, including grammar correction, wording refinement, and summarization of our own text. No model outputs were used to generate new ideas, methods, or experimental results. All technical contributions, algorithms, and experimental analyses were designed, implemented, and validated by the authors without LLM involvement

## A.2 MOTIVATION.

Visual representations must meet two inherently competing requirements: on the one hand, long-range context and global scale consistency (low-frequency content) are crucial for robust classification, object saliency, and within-region coherence; on the other hand, geometric discontinuities at object and occlusion boundaries (high-frequency content) must be strictly preserved to avoid edge fattening and cross-category leakage. Common practice separates these needs into a global module (attention/large-kernel conv) and a boundary module (TV/guided filtering), arranged in parallel or series, but lacking a unified generative principle. This often leads to (i) cancellation and time-scale drift between global and local effects, (ii) parameter non-identifiability (different strength combinations yield similar appearances), and (iii) non-physical propagation near boundaries. We therefore unify feature propagation in the backbone under a single diffusion semigroup. In the spectral domain, we adopt a heat-kernel multiplier to realize a one-shot, interpretable, contractive global low-pass; in the spatial domain, we adopt anisotropic diffusion that propagates along boundary tangents while suppressing the normal direction, thereby preserving geometric jumps. Both are driven by a shared latent control field that jointly parameterizes the spectral attenuation $k(\omega)$ and the diffusion tensor $D(x)$, aligning the evidence that decides "which frequencies to suppress" and "along which directions to diffuse." On the operator level, we impose a PDE residual (strong form) and an energy consistency (weak form) to ensure that the two steps are legitimate approximations of the same generator, and we regularize boundary behavior via a soft zero-normal-flux condition. This design provides a unified, stable, and interpretable structural prior—achieving global consistency plus boundary fidelity across tasks—without heavy parameterization or high complexity, yielding better scale consistency, boundary quality, and training controllability.

**Why a heat-kernel multiplier in the spectral domain and anisotropic diffusion in the spatial domain?** We decompose the diffusion operator into a global, constant-coefficient part and a local, spatially varying part and approximate the same semigroup $e^{tL}$ via Lie–Trotter/Strang splitting:

$$L = L_g + L_\ell, \quad L_g u = \nabla \cdot (\bar{D} \nabla u), \qquad L_\ell u = \nabla \cdot \big( (D(x) - \bar{D}) \nabla u \big), \tag{17}$$

where $\bar{D}$ is the spatial average of $D(x)$. The constant-coefficient component $L_g$ is diagonalized in the DCT basis and admits a closed-form temporal evolution given by the heat-kernel multiplier $e^{-t_g \Lambda(\omega) k(\omega)}$, enabling a one-shot, globally stable low-pass (an $L_2$-contractive semigroup with additive time) at $O(N \log N)$ cost and supporting directional attenuation. In contrast, the spatially varying anisotropic component $L_\ell$ is not shift-invariant and cannot be represented by a single global spectral multiplier; it must be realized in the pixel domain via local flux updates to enforce tangential pass / normal suppression and naturally handle zero-flux boundary conditions under standard CFL stability. If one attempted to model the spatially varying part in the spectral domain, it would require non-shift-invariant, data-dependent spectral operators that are expensive and hard to train stably; if one relied solely on spatial diffusion for global consistency, many iterations would be needed to approximate the heat kernel and time-scale drift easily emerges. Hence, the division "heat kernel for global consistency, anisotropic diffusion for local boundary control" is theoretically coherent (same generator via splitting), numerically stable, and practically efficient.

## A.3 WHY START FROM HEAT CONDUCTION: AN ISOMORPHISM BETWEEN PHYSICAL−MATHEMATICAL PROPERTIES AND VISUAL NEEDS

Heat conduction (anisotropic diffusion) is not only a physical concept, but also an operator that satisfies a set of axiomatic properties crucial to vision. These properties make it a unified tool from low-level filtering to high-level geometric reconstruction.

**(P1) Semigroup and scale space.** The heat kernel $K_t$ satisfies the convolution semigroup

$$K_{t_1} * K_{t_2} \; = \; K_{t_1+t_2}, \qquad \widehat{K_t}(\omega) = \exp\big( - t\, \omega^\top D\, \omega \big), \tag{18}$$

with the evolution $u(t) = K_t * u_0 = e^{tL} u_0$. The parameter $t$ plays the roles of both scale (low→high frequency) and time (short→long diffusion), naturally forming a causal scale space: as $t$ increases, no new structures are introduced—only progressive attenuation of high frequencies.

**(P2) Energy dissipation and nonexpansiveness (Stability).** Diffusion is the steepest-descent flow of the Dirichlet energy:

$$\mathcal{E}(u) = \tfrac{1}{2} \int (\nabla u)^\top D(\mathbf{x})\, \nabla u\, d\mathbf{x}, \quad \frac{d}{dt}\mathcal{E}(u(t)) = -\int (\nabla u)^\top D\, \nabla u \;\le 0. \tag{19}$$

As long as $D(\mathbf{x}) \succeq 0$, the operator $e^{tL}$ is $L_2$-nonexpansive (does not amplify energy/noise). This provides theoretical guarantees for numerical stability and controllability in end-to-end learning.

**(P3) Maximum principle and artifact-free behavior.** Isotropic or weakly anisotropic diffusion satisfies a (discrete) maximum principle: no new interior extrema are created. Hence it does not introduce ringing artifacts or spurious steps, which helps preserve depth monotonicity and hierarchical structure.

**(P4) Variational–statistical consistency (Regularization & MAP).** For any $t > 0$,

$$u(t) \; = \; \arg\min_u \; \frac{1}{2t}\, \|u - u_0\|_2^2 \; + \; \mathcal{E}(u), \tag{20}$$

i.e., one diffusion step is equivalent to the closed-form subproblem of Tikhonov regularization. In probabilistic graphical models, $p(u) \propto \exp\{-\mathcal{E}(u)/\sigma^2\}$ gives a Gaussian MRF prior, and diffusion corresponds to its MAP smoothing.

**(P5) Anisotropy and geometric alignment (Geometry-aware).** When

$$D(\mathbf{x}) = R(\theta)\, \mathrm{diag}(\kappa_\parallel, \kappa_\perp)\, R(\theta)^\top, \qquad 0 \le \kappa_\perp \le \kappa_\parallel, \tag{21}$$

diffusion proceeds tangentially along boundaries while being suppressed in the normal direction ($\kappa_\perp \downarrow$), achieving boundary-faithful controlled propagation. When $\theta$ is driven by a structure tensor or a latent control field, the diffusion direction aligns adaptively with image geometry, avoiding cross-instance "bleeding."

**(P6) Frequency–space duality (Computational duality).** Under a constant-coefficient approximation, the heat kernel is strictly low-pass in the frequency domain, $\exp(-t\, \omega^\top D\, \omega)$, and an anisotropic Gaussian in the spatial domain, $G_{t,D}(\mathbf{x}) \propto \exp(-\frac{1}{4t}\mathbf{x}^\top D^{-1}\mathbf{x})$. This allows us to perform **global consensus** via one-shot stable low-pass in the frequency domain, and **boundary-faithful** propagation via explicit small steps in the spatial domain; both are induced by $e^{tL}$ and are merely implementations in different coordinate systems.

Principled extension to vision: By (P1) and (P6), a single frequency-domain heat-kernel step robustly aggregates low-frequency and long-range context, mitigating scale drift and holes; by (P5), spatial anisotropic diffusion suppresses normal-to-edge flow while allowing tangential propagation, preserving true discontinuities and avoiding edge fattening; by (P2) and (P4), diffusion unifies gradient-flow energy dissipation and Tikhonov-style regularization, enabling numerically stable, end-to-end learning jointly with photometric/geometric data terms; by (P3) it creates no new extrema and reduces ringing, and (P4)/(P6) provide clear semantics— t acts as the scale and D encodes the anisotropic geometry—yielding an artifact-free, interpretable, and easily visualizable formulation.

## A.4   Proof of Spectral and Spatial Approximations of the Diffusion Generator

This subsection rigorously clarifies that our "spectral one-step" and "spatial explicit step" are not heuristic concatenations but two learnable discretizations of the same diffusion generator $e^{tL}$. We further establish the validity and error control of Lie–Trotter/Strang splitting. This common origin guarantees that the composite mapping inherits energy decay and $L_2$-nonexpansiveness during training and inference.

**Setup and notation.** On the discrete grid $\Omega = \{1, \dots, H\} \times \{1, \dots, W\}$ with Neumann (reflection) boundary conditions, define

$$Lu \triangleq \nabla \cdot \big( D(\mathbf{x}) \nabla u \big), \qquad \langle f, g \rangle \triangleq \sum_{\mathbf{x}} f(\mathbf{x}) \, g(\mathbf{x}),$$

where $D(\mathbf{x}) \in \mathbb{R}^{2 \times 2}$ is symmetric positive definite (SPD). The discrete Green identity yields $\langle u, Lu \rangle = -\langle \nabla u, D \nabla u \rangle \leq 0$, so $L$ is negative semidefinite and the diffusion energy $\mathcal{E}(u) = \frac{1}{2} \langle \nabla u, D \nabla u \rangle$ is monotonically non-increasing in time.

### A.4.1 SPECTRAL–TEMPORAL EQUIVALENCE: CONSTANT-COEFFICIENT CASE

If $D(\mathbf{x}) \equiv D_0 \succeq 0$ is constant and $L_0 u = \nabla \cdot (D_0 \nabla u)$, then under the 2D DCT basis (Neumann boundary) there exist an orthogonal matrix $Q$ and a nonnegative diagonal spectrum $\Lambda = \mathrm{diag}(\lambda_\omega)$ such that $L_0 = Q \, (-\Lambda) \, Q^\top$. Hence

$$e^{tL_0} = Q \, e^{-t\Lambda} \, Q^\top, \quad \text{i.e.,} \quad \widehat{u}(\omega, t) = e^{-t\lambda_\omega} \, \widehat{u}_0(\omega).$$

Therefore, the temporal evolution $u(t) = e^{tL_0} u(0)$ is numerically equivalent to exponential attenuation per frequency mode.

**Proof sketch:** With constant coefficients, $L_0$ is an elliptic difference operator; under Neumann boundary conditions it is diagonalized by the DCT. The diagonal entries are eigenvalues $\{\lambda_\omega\} \geq 0$. Exponentiating the diagonal form gives $e^{tL_0} = Q \, e^{-t\Lambda} \, Q^\top$, i.e., mode-wise exponential decay.

**corollary:** Identifiability of spectral kernel and time step: With a learnable spectral kernel $M(\omega) = \exp\{-t_g \Lambda(\omega) k(\omega)\}$, $k(\omega) \geq 0$, if $k(\omega) \equiv 1$ then $M(\omega)$ exactly corresponds to $e^{t_g L_0}$ with step $t_g$; if $k$ is a low-order radial/angular perturbation, then $M$ approximates the symbol of $e^{t_g (L_0 + \delta L)}$. Thus $k$ and $t_g$ together determine the time scale and anisotropy strength.

**proposition:** First-order consistency and stability of explicit Euler: The explicit Euler step $u^+ = u + \Delta t \, Lu$ is a first-order approximation to $e^{\Delta t L}$ with local truncation error $\mathcal{O}(\Delta t^2)$. If $\Delta t \leq c / \lambda_{\max}(-L)$ (for the 5-point stencil a conservative choice is $c \approx \frac{1}{4}$), then $\|u^+\|_2 \leq \|u\|_2$ and $\mathcal{E}(u^+) \leq \mathcal{E}(u)$.

**Proof sketch** From $e^{\Delta t L} = I + \Delta t L + \mathcal{O}(\Delta t^2)$ we get consistency; stability follows from the spectral radius bound and the energy inequality (CFL condition).

**proposition:** Spectral one-step $\approx$ many tiny explicit steps (Chernoff/product formula)] For any nonnegative diagonal spectrum $\Lambda$ and $k(\omega) \geq 0$,

$$e^{-t \Lambda k} = \lim_{m \to \infty} \left( I - \frac{t}{m} \Lambda k \right)^m.$$

That is, a single closed-form spectral propagation equals the limit of $m \to \infty$ explicit Euler steps with size $t/m$.

**Proof sketch** Use the scalar limit $e^{-\alpha t} = \lim_m (1 - \frac{\alpha t}{m})^m$ elementwise on the diagonal; apply similarity transforms back to the spatial domain.

### A.4.2 VARIABLE-COEFFICIENT CASE AND SPLITTING VALIDITY.

When $D(\mathbf{x})$ varies spatially, write $L = L_g + L_\ell$, where $L_g$ captures global, shift-invariant constant-coefficient effects (or their spectral surrogate) and $L_\ell$ captures local, spatially varying anisotropy; both are negative semidefinite.

**theorem:**Lie–Trotter/Strang splitting and error bounds For any $t \geq 0$,

$$e^{t(L_g + L_\ell)} = e^{tL_g} e^{tL_\ell} + \mathcal{O}(t^2), \qquad e^{t(L_g + L_\ell)} = e^{\frac{t}{2} L_g} e^{tL_\ell} e^{\frac{t}{2} L_g} + \mathcal{O}(t^3).$$

Moreover, there exist constants $C_2, C_3$ depending on the commutator $[L_g, L_\ell]$ such that

$$\left\| e^{t(L_g + L_\ell)} - e^{tL_g} e^{tL_\ell} \right\|_{2 \to 2} \leq C_2 \, t^2 \, \|[L_g, L_\ell]\|_{2 \to 2} + \mathcal{O}(t^3).$$

**Proof sketch** Apply the Baker–Campbell–Hausdorff expansion and semigroup continuity to obtain first/second-order errors; the commutator controls the leading noncommutative error term.

**corollary:** Transfer of contraction and composite stability]: If the single-step operators $T_{\text{spec}} = e^{t_g L_g}$ and $T_{\text{space}} = I + \Delta t\, L_\ell$ satisfy $\|T_{\text{spec}}\|_{2\to 2} \leq 1$ and $\|T_{\text{space}}\|_{2\to 2} \leq 1$ (respectively from $|e^{-t_g \lambda}| \leq 1$ and CFL/energy decay), then the composite mapping $T_{\text{space}} \circ T_{\text{spec}}$ is also $L_2$-nonexpansive and energy non-increasing: $\|u^+ - v^+\|_2 \leq \|u - v\|_2$, $\mathcal{E}(u^+) \leq \mathcal{E}(\tilde{u}) \leq \mathcal{E}(u)$.

**Why spectral $\leftrightarrow$ temporal evolution are "the same thing".** Theorem A.4.1 shows that when coefficients are frozen (or locally/globally averaged), the temporal evolution $u(t) = e^{tL}$ is mode-wise exponential decay in the spectral domain; Proposition A.4.1 further shows that a single spectral closed-form step equals the composite limit of infinitely many tiny explicit time steps. For variable coefficients, Theorem A.4.2 proves that "spectral-then-spatial" (and optionally Strang) is a legitimate approximation of $e^{tL}$, with an error controlled by the commutator; Corollary A.4.2 transfers single-step contraction to the composite, ensuring overall stability. Thus, "spectral" and "temporal" are not competing mechanisms but two coordinate representations and propagation modes under the same diffusion semigroup—the former provides a single-pass, globally stable low-pass propagation, while the latter offers local, boundary-aligned refinement steps.

## A.5   GREEN'S IDENTITY

**Energy and stability via a discrete Green identity.** We formalize the energy dissipation and stability of our spatial diffusion step using the (discrete) Green identity, i.e., a summation–by–parts relation that mirrors integration by parts for PDEs.

**Setting.** Let $u \in \mathbb{R}^{H \times W}$ be a feature map on a regular grid with spacings $\Delta x = \Delta y = 1$. The anisotropic diffusion operator is discretized as

$$L_h u \triangleq \nabla_h \cdot (D\, \nabla_h u), \tag{22}$$

where $D(x)$ is a symmetric positive semidefinite diffusion tensor, $\nabla_h$ and $\nabla_h\cdot$ are the standard forward–difference gradient and flux–difference divergence with Neumann (reflective) boundary so that the normal flux vanishes on $\partial\Omega$.

**Discrete energy.** Define the diffusion energy

$$\mathcal{E}_h(u) = \tfrac{1}{2} \langle \nabla_h u,\, D\, \nabla_h u \rangle_h, \tag{23}$$

where $\langle \cdot, \cdot \rangle_h$ is the discrete inner product (sum over pixels). Under Neumann boundaries, the discrete Green identity (summation by parts) states

$$\langle \nabla_h u,\, D\, \nabla_h v \rangle_h = -\langle u,\, \nabla_h \cdot (D\, \nabla_h v) \rangle_h + \underbrace{\langle u,\, (D\nabla_h v) \cdot n \rangle_{\partial\Omega}}_{=\,0}, \tag{24}$$

so the boundary term vanishes and we obtain the operator–energy pairing

$$\mathcal{E}_h(u) = \tfrac{1}{2} \langle u,\, -L_h u \rangle_h. \tag{25}$$

Eq. equation 25 implies that $-L_h$ is symmetric positive semidefinite (SPSD) with respect to $\langle \cdot, \cdot \rangle_h$, i.e., $L_h$ is *dissipative*.

**Explicit step and CFL.** Consider the explicit Euler update

$$u^{k+1} = u^k - \Delta t\, L_h u^k. \tag{26}$$

Using equation 25 and the SPSD property of $-L_h$,

$$\mathcal{E}_h(u^{k+1}) - \mathcal{E}_h(u^k) \leq -\Delta t\, \langle L_h u^k,\, L_h u^k \rangle_h + \tfrac{1}{2}(\Delta t)^2 \langle L_h u^k,\, (-L_h)\, L_h u^k \rangle_h, \tag{27}$$

which is nonpositive provided $\Delta t$ is below a Courant–Friedrichs–Lewy (CFL) bound that depends on the maximal face conductances (cf. our implementation). Hence, for $\Delta t \leq \Delta t_{\text{CFL}}$, the scheme is energy diminishing and $L_2$-nonexpansive.

**Interpretation.** The discrete Green identity is the algebraic backbone that (i) links the gradient-form energy $\|\nabla_h u\|_D^2$ to the divergence-form operator $L_h$, (ii) establishes the dissipativity and symmetry of the discrete diffusion, and (iii) yields a rigorous, transparent stability condition (CFL) under reflective boundaries. This furnishes a physics-consistent justification for our energy loss and the choice of time step in training/inference.

### A.6 Spectral and Spatial Head Discretization under a Unified Diffusion View

#### A.6.1 Discretization of the Spectral Attenuation Head

To instantiate Equ. 9 under the heat–conduction framework, we take the spectral multiplier $M(\omega)$ to be a heat kernel and provide both the general Fourier form and a DCT-based discrete implementation.

**General Fourier form (periodic boundary):** For isotropic heat conduction, the spectral response

$$M(\omega) = \exp\big[-t\,\kappa\,\|\omega\|^2\big] = \exp\big[-t\,\kappa(\omega_x^2 + \omega_y^2)\big], \tag{28}$$

where $t > 0$ is the diffusion time and $\kappa > 0$ is the diffusion coefficient. Substituting back yields

$$\tilde{u} = \mathcal{F}^{-1}\Big(\exp[-t\kappa\|\omega\|^2] \odot \mathcal{F}(u)\Big), \tag{29}$$

**Learnable heat kernel (spectral parameterization).** To obtain content adaptivity, we replace the constant $\kappa$ with a nonnegative, learnable weight $k(\omega) \geq 0$, and use the symbol of the discrete Laplacian $\Lambda(\omega)$ to express the quadratic frequency form:

$$\tilde{u} = \mathcal{F}^{-1}\Big(\exp[-t_g\,\Lambda(\omega)\,k(\omega)] \odot \mathcal{F}(u)\Big), \qquad t_g > 0. \tag{30}$$

Here $k(\omega)$ can be parameterized with a radial + second-order angular basis to encode directional low-pass behavior while ensuring dissipativity $0 < M(\omega) \leq 1$.

**DCT discrete implementation (Neumann/reflective boundary).** Let $\mathcal{F}$ be the 2D DCT/IDCT. The discrete Laplacian is diagonalized in the DCT basis with eigen-spectrum

$$\Lambda_{p,q} = 4\sin^2\Big(\frac{\pi p}{2H}\Big) + 4\sin^2\Big(\frac{\pi q}{2W}\Big), \qquad p = 0, \ldots, H-1, \ \ q = 0, \ldots, W-1. \tag{31}$$

Construct the spectral multiplier $M[p,q] = \exp\big[-t_g\,\Lambda_{p,q}\,k_{p,q}\big]$, and compute

$$\tilde{u} = \mathrm{IDCT}_{2D}\Big(M[p,q] \odot \mathrm{DCT}_{2D}(u)\Big). \tag{32}$$

This implementation matches reflective boundary conditions and is well suited for natural images; each channel can be processed independently, with complexity $O(N \log N)$.

#### A.6.2 Discretization of the spectral attenuation head

We discretize Equ. 11 on a regular pixel grid $\Omega = \{(i,j) \mid i = 0, \ldots, H-1, \ j = 0, \ldots, W-1\}$. Unless otherwise stated, pixel spacings are $\Delta x = \Delta y = 1$. Neumann (reflective) boundary conditions are used. **Tensor parameters and directional projection.** At each pixel we define the anisotropic diffusion tensor

$$D_{i,j} = R(\theta_{i,j})\,\mathrm{diag}\big(\kappa_{\|,i,j},\,\kappa_{\perp,i,j}\big)\,R(\theta_{i,j})^\top, \qquad \kappa_\perp \geq 0, \ \ \kappa_\| \geq \kappa_\perp. \tag{33}$$

Its effective conductivities in the horizontal/vertical axes are

$$c_x(i,j) = \kappa_\| \cos^2\theta + \kappa_\perp \sin^2\theta, \qquad c_y(i,j) = \kappa_\| \sin^2\theta + \kappa_\perp \cos^2\theta. \tag{34}$$

**Face conductances (finite volume / harmonic mean).** To ensure symmetry and dissipation, the conductances on the four faces are set by the harmonic mean:

$$\begin{aligned}
w_E(i,j) &= \frac{2\,c_x(i,j)\,c_x(i{+}1,j)}{c_x(i,j) + c_x(i{+}1,j) + \varepsilon}, \quad w_W(i,j) = \frac{2\,c_x(i{-}1,j)\,c_x(i,j)}{c_x(i{-}1,j) + c_x(i,j) + \varepsilon}, \\
w_N(i,j) &= \frac{2\,c_y(i,j)\,c_y(i,j{+}1)}{c_y(i,j) + c_y(i,j{+}1) + \varepsilon}, \quad w_S(i,j) = \frac{2\,c_y(i,j{-}1)\,c_y(i,j)}{c_y(i,j{-}1) + c_y(i,j) + \varepsilon}.
\end{aligned} \tag{35}$$

**Discrete divergence (one explicit step).** Finite-volume flux-difference approximation divergence,

$$\Big[\nabla\cdot(D\nabla\tilde{u})\Big]_{i,j} = \frac{F_E - F_W}{\Delta x} + \frac{F_N - F_S}{\Delta y}, \tag{36}$$

$$F_E = \frac{w_E(i,j)}{\Delta x}\big(\tilde{u}_{i+1,j} - \tilde{u}_{i,j}\big), \quad F_W = \frac{w_W(i,j)}{\Delta x}\big(\tilde{u}_{i,j} - \tilde{u}_{i-1,j}\big),$$
$$F_N = \frac{w_N(i,j)}{\Delta y}\big(\tilde{u}_{i,j+1} - \tilde{u}_{i,j}\big), \quad F_S = \frac{w_S(i,j)}{\Delta y}\big(\tilde{u}_{i,j} - \tilde{u}_{i,j-1}\big), \tag{37}$$

the explicit Euler update reads

$$u_{i,j}^+ = \tilde{u}_{i,j} + \Delta t\left[\frac{w_E(\tilde{u}_{i+1,j} - \tilde{u}_{i,j}) - w_W(\tilde{u}_{i,j} - \tilde{u}_{i-1,j})}{(\Delta x)^2} + \frac{w_N(\tilde{u}_{i,j+1} - \tilde{u}_{i,j}) - w_S(\tilde{u}_{i,j} - \tilde{u}_{i,j-1})}{(\Delta y)^2}\right].$$
$$\tag{38}$$

For $\Delta x = \Delta y = 1$ this simplifies to the four-neighbor form:

$$u_{i,j}^+ = \tilde{u}_{i,j} + \Delta t\Big[w_E(\tilde{u}_{i+1,j} - \tilde{u}_{i,j}) - w_W(\tilde{u}_{i,j} - \tilde{u}_{i-1,j}) + w_N(\tilde{u}_{i,j+1} - \tilde{u}_{i,j}) - w_S(\tilde{u}_{i,j} - \tilde{u}_{i,j-1})\Big].$$
$$\tag{39}$$

**Stability (CFL) and properties.** To guarantee energy decay and $L_2$-nonexpansiveness for the explicit step, choose

$$\Delta t \leq \left(\max_{i,j}\left\{\frac{w_E + w_W}{(\Delta x)^2} + \frac{w_N + w_S}{(\Delta y)^2}\right\}\right)^{-1}. \tag{40}$$

**Implementation note.** Neumann boundaries are implemented by reflection padding; $\varepsilon > 0$ avoids division by zero in the harmonic mean. The scheme consists only of per-pixel algebra and four-neighborhood differences and can be implemented as a fully differentiable network layer.

### A.7 COMPLEXITY ANALYSIS: SPECTRAL AND SPATIAL OPERATORS

We analyze the per-layer time complexity for inputs of size $u \in \mathbb{R}^{B \times C \times H \times W}$. Let $N \triangleq HW$ be the number of pixels per channel.

#### A.7.1 SPECTRAL OPERATOR: HEAT-KERNEL MULTIPLIER.

The spectral step is

$$\tilde{u} = \mathrm{IDCT}_{2D}\Big(M(\omega) \odot \mathrm{DCT}_{2D}(u)\Big), \qquad M(\omega) = \exp\big[-t_g \Lambda(\omega) k(\omega)\big]. \tag{41}$$

Definition: one spectral step consists of a 2D DCT, a pointwise frequency multiplication with the heat-kernel multiplier $M(\omega)$, and a 2D IDCT. The computation of $M(\omega)$ is a per-frequency Hadamard product, linear in the number of pixels.

**Cost of separable 2D DCT/IDCT.** Assume a fast 1D DCT of length $n$ whose cost is at most $a\,n\log_2 n + b\,n$ (constants $a, b > 0$). A separable 2D DCT/IDCT on an $H \times W$ array is realized by $H$ row-wise 1D transforms of length $W$ and $W$ column-wise 1D transforms of length $H$. Hence the total cost is bounded by

$$a\,HW\,(\log_2 H + \log_2 W) + b\,HW \;=\; \Theta\big(HW(\log H + \log W)\big). \tag{42}$$

**Conclusion (spectral step).** The per-channel, per-sample cost of one spectral step is

$$T_{\mathrm{spec}}(H, W) = \Theta\big(N(\log H + \log W)\big). \tag{43}$$

For a batch of size $B$ and $C$ channels,

$$T_{\mathrm{spec}} = \Theta\big(B\,C\,H\,W(\log H + \log W)\big) = \Theta(BC\,N\log N). \tag{44}$$

Justification: one 2D DCT and one 2D IDCT each cost $\Theta(HW(\log H + \log W))$; the intermediate Hadamard product is $O(HW)$ and negligible relative to the transforms. Multiplying by $B\,C$ yields the result; since $\log H + \log W = \Theta(\log(HW)) = \Theta(\log N)$, we obtain $\Theta(BC\,N\log N)$.

**Variants.** (i) Block DCT: with non-overlapping blocks of size $m \times m$ (constant $m$), $T_{\mathrm{spec}} = \Theta(BC\,N\log m) = \Theta(BC\,N)$. (ii) Polynomial realization: with a degree-$R$ Chebyshev/rational approximation $M(\omega) \approx \sum_{r=0}^{R} a_r[\Lambda(\omega)]^r$, each power corresponds to one sparse Laplacian application, thus $T_{\mathrm{spec}} = \Theta(BC\,R\,N)$.

A.7.2 SPATIAL OPERATOR: ONE ANISOTROPIC DIFFUSION STEP.

The explicit Euler update is

$$u^+ = \tilde{u} + \Delta t \, \nabla \cdot \big(D(x) \, \nabla \tilde{u}\big), \qquad D(x) = R(\theta) \begin{bmatrix} \kappa_\parallel & 0 \\ 0 & \kappa_\perp \end{bmatrix} R(\theta)^\top. \tag{45}$$

We discretize with fixed stencils (4/8-neighborhood finite differences or finite volumes with harmonic edge weights) under Neumann boundary conditions, computing the gradient $\nabla$, the flux $J = D\nabla\tilde{u}$, and the divergence $\nabla \cdot J$.

**Per-pixel constant work.** With a fixed-size stencil (e.g., 4 or 8 neighbors), centered differences, a $2 \times 2$ matrix–vector product for the flux, and divergence evaluation each require $O(1)$ arithmetic operations per pixel. **Conclusion (spatial step).** A single explicit anisotropic diffusion step over $B$ samples and $C$ channels costs

$$T_{\text{space-step}} = \Theta(N), \qquad T_{\text{space}} = \Theta(BC\,N). \tag{46}$$

If $S$ explicit substeps are performed for stronger anisotropy or improved stability,

$$T_{\text{space}}(S) = \Theta(BC\,S\,N). \tag{47}$$

Assembly cost: predicting $\theta, \kappa_\parallel, \kappa_\perp$ from the control field $g(x)$ via a $1{\times}1$ (or shallow) convolutional head and enforcing SPD via $\kappa_\perp = \text{softplus}(a)$, $\kappa_\parallel = \kappa_\perp + \text{softplus}(b)$ are pointwise operations with total cost $\Theta(BC\,N)$, hence do not change the order above. Stability: the CFL condition $\Delta t \leq 1/\big(4\max_x \kappa_\parallel(x) + \varepsilon\big)$ constrains the step size but does not affect the complexity order.

A.7.3 COMPARISON WITH SELF-ATTENTION.

Global self-attention over $N$ tokens has per-layer complexity

$$T_{\text{attn}} = \Theta(B\,C\,N^2). \tag{48}$$

In contrast, one spectral step plus one spatial step costs

$$T_{\text{spec}} + T_{\text{space}} = \Theta(BC\,N\log N) + \Theta(BC\,N), \tag{49}$$

which is asymptotically smaller for large $N$. Windowed attention of size $w \times w$ scales as $\Theta(BC\,N\,w^2)$ and becomes near-linear for small $w$, at the expense of losing global coupling that our spectral step preserves.

**Memory/bandwidth and parallelism.** The main memory terms are feature tensors and intermediate gradient/flux buffers, both $\Theta(BC\,N)$. DCT/IDCT can employ in-place/blockwise implementations and streaming buffers; the spatial step accesses constant-size neighborhoods, favoring multi-core and tensor-core parallelization.

A.8 ARCHITECTURE IMPLEMENTATION

| Size | Tiny | Small | Base |
|---|---|---|---|
| Stem | 3×3 conv with stride 2; Norm; GELU; 3×3 conv with stride 2; Norm | | |
| Downsampling | 3×3 conv with stride 2; Norm | | |
| MLP ratio | 4 | | |
| Classifier head | Global average pooling, Norm, MLP | | |
| Layers | (2,2,6,2) (classification) | (2,2,18,2) (classification) | (2,2,18,2) (segmentation) |
| | (2,2,5,2) (others) | (2,2,16,2) (others) | (4,4,20,4) (others) |
| Channels | (96,192,384,768) | (96,192,384,768) | (128,256,512,1024) (segmentation) |
| | | | (96,192,384,768) (others) |

Table 3: Configurations of UniDiff . The contents in the tuples represent configurations for four stages.

Given an input image of spatial resolution $H \times W$, **UniDiff** first employs a stem module to partition the image into patches, producing a 2D feature map of resolution $H/4 \times W/4$. We then build a

hierarchical representation with progressively reduced resolutions—$H/4 \times W/4$, $H/8 \times W/8$, and $H/16 \times W/16$—and correspondingly increased channel widths. Each stage consists of a *downsampling* layer followed by several **S2D** blocks (with the **first stage** as an exception: it starts directly with S2D blocks without a preceding downsampling layer).

## A.9 EXPERIMENTAL RESULTS

| Backbone | Mask R-CNN on COCO | | | | UperNet on ADE20K | | |
|---|---|---|---|---|---|---|---|
| | $AP^b$ | $AP^m$ | FPS(images/s) | FLOPs | mIoU | FPS (images/s) | FLOPs |
| Swin-TLiu et al. (2021) | 42.7 | 39.3 | 26.3 | 267G | 44.4 | 31.8 | 237G |
| ConvNeXt-TLiu et al. (2022) | 44.2 | 40.1 | 29.3 | **262G** | 46.0 | **37.8** | **235G** |
| vHeat-T Wang et al. (2025b) | 45.1 | 41.2 | **32.7** | 272G | 46.9 | 36.7 | 235G |
| GroupMamba-T Shaker et al. (2025) | 47.6 | 42.9 | – | 279G | 48.6 | – | 279G |
| TransNeXt-T Shi (2024) | 49.9 | 45.6 | 25.2 | – | 51.1 | 28 | – |
| Ours | **50.2** | **45.9** | 31.5 | 278G | **51.4** | 35.8 | **239G** |
| Swin-SLiu et al. (2021) | 44.8 | 40.9 | 19.7 | 359G | 47.6 | 22.1 | 261G |
| ConvNeXt-SLiu et al. (2022) | 45.4 | 41.8 | 20.2 | 349G | 48.7 | **27.7** | 257G |
| vHeat-S Wang et al. (2025b) | 46.8 | 42.3 | **25.9** | **348G** | 49.1 | 26.1 | **254G** |
| BiFormer-S Zhu et al. (2023) | 47.8 | 43.2 | – | 352 | 49.8 | – | 257 |
| TransNeXt-S Shi (2024) | 51.1 | 45.5 | 18.2 | – | 52.2 | 21 | – |
| Ours | **51.3** | **45.8** | 25.2 | 353G | **52.4** | 25.5 | 258G |
| Swin-BLiu et al. (2021) | 46.9 | 42.3 | 13.8 | 504G | 48.1 | 19.2 | 299G |
| ConvNeXt-BLiu et al. (2022) | 47.0 | 42.7 | 14.1 | 486G | 49.1 | 21.6 | **293G** |
| vHeat-BWang et al. (2025b) | 47.7 | 43.0 | **20.2** | **432G** | 49.6 | **23.6** | 293G |
| metaformeYu et al. (2024a) | 48.6 | 43.7 | 18 | – | 51.7 | 9.8 | – |
| BiFormer-B Zhu et al. (2023) | 48.6 | 43.7 | – | – | 51.0 | – | – |
| TransNeXt-S Shi (2024) | 51.1 | 45.9 | 12 | – | 53.0 | 16 | – |
| (Ours) | **51.5** | **46.1** | 19.4 | 437G | **53.2** | 22.8 | 299G |

Table 4: Left—object detection and instance segmentation on COCOLin et al. (2014) (reporting $AP_b$ and $AP_m$; FLOPs computed at 1280×800 input). $AP_b$ and $AP_m$ denote box AP and mask AP, respectively. Right—semantic segmentation on ADE20KZhou et al. (2017) Using UperNetXiao et al. (2018); FLOPs computed at 512×512.

| ImageNet-1KRussakovsky et al. (2015) $224^2$ pre-trained models | | | | | | | | |
|---|---|---|---|---|---|---|---|---|
| Model | #Params. (M) | FLOPs (G) | IN-1K ↑ Top-1(%) | IN-C ↓ mCE(%) | IN-A ↑ Top-1(%) | IN-R ↑ Top-1(%) | Sketch ↑ Top-1(%) | IN-V2 ↑ Top-1(%) |
| PVT-Tiny Wang et al. (2021) | 13.2 | 1.9 | 75.1 | 79.6 | 8.2 | 33.7 | 21.3 | 63.0 |
| PVTv2-B1 Wang et al. (2022) | 14.0 | 2.1 | 78.7 | 62.6 | 14.7 | 41.8 | 28.9 | 66.9 |
| BiFormer-T ? | 13.1 | 2.2 | 81.4 | 55.7 | 25.7 | 45.4 | 31.5 | 70.6 |
| EfficientFormerV2-S2 Li et al. (2022b) | 12.7 | 1.3 | 81.6 | 51.4 | 22.2 | 44.9 | 30.8 | 70.7 |
| TransNeXt-Micro (Ours) | 12.8 | 2.7 | **82.5** | **50.8** | **29.9** | **45.8** | **33.0** | **72.6** |
| DeiT-Small/16 Touvron et al. (2021) | 22.1 | 4.6 | 79.9 | 54.6 | 19.8 | 41.9 | 29.1 | 68.4 |
| Swin-T Liu et al. (2021) | 28.3 | 4.5 | 81.2 | 62.0 | 21.7 | 41.3 | 29.0 | 69.7 |
| PVTv2-B2 Wang et al. (2022) | 25.4 | 4.0 | 82.0 | 52.6 | 27.9 | 45.1 | 32.8 | 71.6 |
| ConvNeXt-T Liu et al. (2022) | 28.6 | 4.5 | 82.1 | 53.2 | 24.2 | 47.2 | 33.8 | 71.0 |
| Focal-T Yang et al. (2021) | 29.1 | 4.9 | 82.2 | – | – | – | – | – |
| FocalNet-T (LRF) Yang et al. (2022) | 28.6 | 4.5 | 82.3 | 55.0 | 23.5 | 45.1 | 31.8 | 71.2 |
| MaxViT-Tiny Tu et al. (2022) | 30.9 | 5.6 | 83.4 | 49.6 | 32.8 | 48.3 | 36.3 | 72.9 |
| BiFormer-S ? | 25.5 | 4.5 | 83.8 | 48.5 | 39.5 | 49.6 | 36.4 | 73.7 |
| TransNeXt-Tiny (Ours) | 28.2 | 5.7 | **84.0** | **46.5** | **39.9** | **49.6** | **37.6** | **73.8** |
| Swin-S Liu et al. (2021) | 49.6 | 8.7 | 83.1 | 54.9 | 32.9 | 44.9 | 32.0 | 72.1 |
| ConvNeXt-S Liu et al. (2022) | 50.2 | 8.7 | 83.1 | 49.5 | 31.3 | 49.6 | 37.1 | 72.5 |
| PVTv2-B3 Wang et al. (2022) | 45.2 | 6.9 | 83.2 | 48.0 | 33.3 | 49.2 | 36.7 | 73.0 |
| Focal-S Yang et al. (2021) | 51.1 | 9.1 | 83.5 | – | – | – | – | – |
| FocalNet-S (LRF) Yang et al. (2022) | 50.3 | 8.7 | 83.5 | 51.0 | 33.8 | 47.7 | 35.1 | 72.7 |
| PVTv2-B4 Wang et al. (2022) | 62.6 | 10.1 | 83.6 | 46.5 | 37.1 | 49.8 | 37.5 | 73.5 |
| BiFormer-B ? | 56.8 | 9.8 | 84.3 | 47.2 | 44.3 | 49.7 | 35.3 | 74.0 |
| MaxViT-Small Tu et al. (2022) | 68.9 | 11.7 | 84.4 | 46.4 | 40.0 | 50.6 | 38.3 | 74.0 |
| TransNeXt-Small (Ours) | 49.7 | 10.3 | **84.7** | **43.9** | **47.1** | **52.5** | **39.7** | **74.8** |
| DeiT-Base/16 Touvron et al. (2021) | 86.6 | 17.6 | 81.8 | 48.5 | 28.1 | 44.7 | 32.0 | 70.9 |
| Swin-B Liu et al. (2021) | 87.8 | 15.4 | 83.5 | 54.5 | 35.9 | 46.6 | 32.4 | 72.3 |
| PVTv2-B5 Wang et al. (2022) | 82.0 | 11.8 | 83.8 | 45.9 | 36.8 | 49.8 | 37.2 | 73.4 |
| Focal-B Yang et al. (2021) | 89.8 | 16.0 | 83.8 | – | – | – | – | – |
| ConvNeXt-B Liu et al. (2022) | 88.6 | 15.4 | 83.8 | 46.8 | 36.7 | 51.3 | 38.2 | 73.7 |
| FocalNet-B (LRF) Yang et al. (2022) | 88.7 | 15.4 | 83.9 | 49.5 | 38.3 | 48.1 | 35.7 | 73.5 |
| TransNeXt-Base (Ours) | 89.7 | 18.4 | **84.8** | **43.5** | **50.6** | **53.9** | **41.4** | **75.1** |
| MaxViT-Base Tu et al. (2022) | 119.5 | 24.0 | 84.9 | 43.6 | 44.2 | 52.5 | 40.1 | 74.5 |

Table 5: A comprehensive comparison on the ImageNet-1KRussakovsky et al. (2015) classification.

| Backbone | Encoder size(M) | #Params. (M) | $AP^b$ | $AP^b_{50}$ | $AP^b_{75}$ | $AP^m$ | $AP^m_{50}$ | $AP^m_{75}$ |
|---|---|---|---|---|---|---|---|---|
| Swin-T Liu et al. (2021) | 28.3 | 47.8 | 43.7 | 66.6 | 47.7 | 39.8 | 63.3 | 42.7 |
| PVTv2-B2 Wang et al. (2022) | 25.4 | 45.3 | 45.3 | 67.1 | 49.6 | 41.2 | 64.2 | 44.4 |
| FocalNet-T (LRF) Yang et al. (2022) | 28.6 | 48.9 | 46.1 | 68.2 | 50.6 | 41.5 | 65.1 | 44.5 |
| Swin-S Liu et al. (2021) | 49.6 | 69.1 | 46.5 | 68.7 | 51.3 | 42.1 | 65.8 | 45.2 |
| CSWin-T Dong et al. (2022) | 23 | 42 | 46.7 | 68.6 | 51.3 | 42.2 | 65.6 | 45.4 |
| Swin-B Liu et al. (2021) | 87.8 | 107.1 | 46.9 | 69.2 | 51.6 | 42.3 | 66.0 | 45.5 |
| PVTv2-B3 Wang et al. (2022) | 45.2 | 64.9 | 47.0 | 68.1 | 51.7 | 42.5 | 65.7 | 45.7 |
| InternImage-T Wang et al. (2023) | 30 | 49 | 47.2 | 69.0 | 52.1 | 42.5 | 66.1 | 45.8 |
| PVTv2-B5 Wang et al. (2022) | 82.0 | 101.6 | 47.4 | 68.6 | 51.9 | 42.5 | 65.7 | 46.0 |
| PVTv2-B4 Wang et al. (2022) | 62.6 | 82.2 | 47.5 | 68.7 | 52.0 | 42.7 | 66.1 | 46.1 |
| InternImage-S Wang et al. (2023) | 50 | 69 | 47.8 | 69.8 | 52.8 | 43.3 | 67.1 | 46.7 |
| SMT-S Tang et al. (2021) | 20.5 | 40.0 | 47.8 | 69.5 | 52.1 | 43.0 | 66.6 | 46.1 |
| BiFormer-S ? | 25.5 | 45.2 | 47.8 | 69.8 | 52.3 | 43.2 | 66.8 | 46.5 |
| CSWin-S Dong et al. (2022) | 35 | 54 | 47.9 | 70.1 | 52.6 | 43.2 | 67.1 | 46.2 |
| FocalNet-S (LRF) Yang et al. (2022) | 50.3 | 72.3 | 48.3 | 70.5 | 53.1 | 43.1 | 67.4 | 46.2 |
| BiFormer-B ? | 56.8 | 76.3 | 48.6 | 70.5 | 53.8 | 43.7 | 67.6 | 47.1 |
| CSWin-B Dong et al. (2022) | 78 | 97 | 48.7 | 70.4 | 53.9 | 43.9 | 67.8 | 47.3 |
| InternImage-B Wang et al. (2023) | 97 | 115 | 48.8 | 70.9 | 54.0 | 44.0 | 67.8 | 47.4 |
| SMT-B Tang et al. (2021) | 32 | 51.7 | 49.0 | 70.2 | 53.7 | 44.0 | 67.6 | 47.4 |
| FocalNet-B (LRF) Yang et al. (2022) | 88.7 | 111.4 | 49.0 | 70.9 | 53.9 | 43.5 | 67.9 | 46.7 |
| TransNeXt-Tiny (Ours) | 28.2 | 47.9 | **49.9** | **71.5** | **54.9** | **44.6** | **68.6** | **48.1** |
| TransNeXt-Small (Ours) | 49.7 | 69.3 | **51.1** | **72.6** | **56.2** | **45.5** | **69.8** | **49.1** |
| TransNeXt-Base (Ours) | 89.7 | 109.2 | **51.7** | **73.2** | **56.9** | **45.9** | **70.5** | **49.7** |

Table 6: Detailed COCOLin et al. (2014) object detection and instance segmentation results using the Mask R-CNN He et al. (2017) 1× schedule, sorted in ascending order based on $AP^b$ performance.

| Model | Encoder size(M) | #Params. (M) | Crop-size | Pre-trained | mIoU (%) | +MS (%) |
|---|---|---|---|---|---|---|
| Swin-T Liu et al. (2021) | 28.3 | 60 | $512^2$ | IN-1K | 44.5 | 45.8 |
| Focal-T Yang et al. (2021) | 29.1 | 62 | $512^2$ | IN-1K | 45.8 | 47.0 |
| ConvNeXt-T Liu et al. (2022) | 28.6 | 60 | $512^2$ | IN-1K | 46.0 | 46.7 |
| FocalNet-T(LRF) Yang et al. (2022) | 28.6 | 61 | $512^2$ | IN-1K | 46.8 | 47.8 |
| Swin-S Liu et al. (2021) | 49.6 | 81 | $512^2$ | IN-1K | 47.6 | 49.5 |
| UniFormer-S Li et al. (2022a) | 22 | 52 | $512^2$ | IN-1K | 47.6 | 48.5 |
| Focal-S Yang et al. (2021) | 51.1 | 85 | $512^2$ | IN-1K | 48.0 | 50.0 |
| Swin-B Liu et al. (2021) | 87.8 | 121 | $512^2$ | IN-1K | 48.1 | 49.7 |
| ConvNeXt-S Liu et al. (2022) | 50.2 | 82 | $512^2$ | IN-1K | 48.7 | 49.6 |
| Focal-B Yang et al. (2021) | 89.8 | 126 | $512^2$ | IN-1K | 49.0 | 50.5 |
| FocalNet-S(LRF) Yang et al. (2022) | 50.3 | 84 | $512^2$ | IN-1K | 49.1 | 50.1 |
| ConvNeXt-B Liu et al. (2022) | 88.6 | 122 | $512^2$ | IN-1K | 49.1 | 49.9 |
| SMT-S Tang et al. (2021) | 20.5 | 50.1 | $512^2$ | IN-1K | 49.2 | 50.2 |
| SMT-B Tang et al. (2021) | 32 | 61.8 | $512^2$ | IN-1K | 49.6 | 50.6 |
| UniFormer-B Li et al. (2022a) | 49.8 | 80 | $512^2$ | IN-1K | 50.0 | 50.8 |
| FocalNet-B(LRF) Yang et al. (2022) | 88.7 | 126 | $512^2$ | IN-1K | 50.5 | 51.4 |
| TransNeXt-Tiny (Ours) | 28.2 | 59 | $512^2$ | IN-1K | **51.1** | **51.5/51.7** |
| TransNeXt-Small (Ours) | 49.7 | 80 | $512^2$ | IN-1K | **52.2** | **52.5/52.8** |
| ConvNeXt-B Liu et al. (2022) | 88.6 | 122 | $640^2$ | IN-22K | 52.6 | 53.1 |
| TransNeXt-Base (Ours) | 89.7 | 121 | $512^2$ | IN-1K | **53.0** | **53.5/53.7** |

Table 7: A comprehensive comparison of semantic segmentation results on the ADE20K datasetZhou et al. (2017) using the UperNetXiao et al. (2018) method. +MS denotes evaluation with multi-scale and flip augmentations.

## A.10 ABLATION STUDY

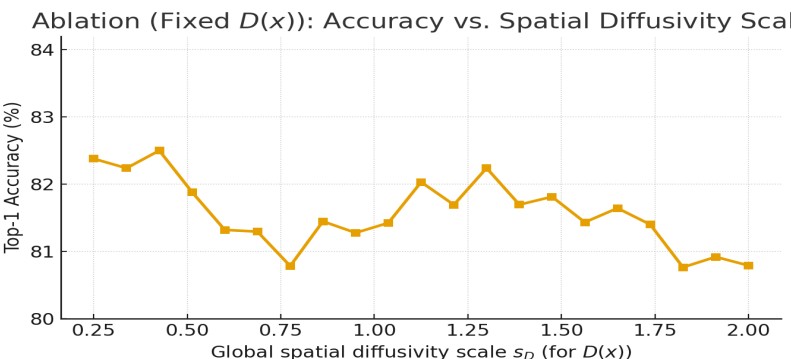

Figure 3: Ablation on spatial diffusivity $D(x)$.

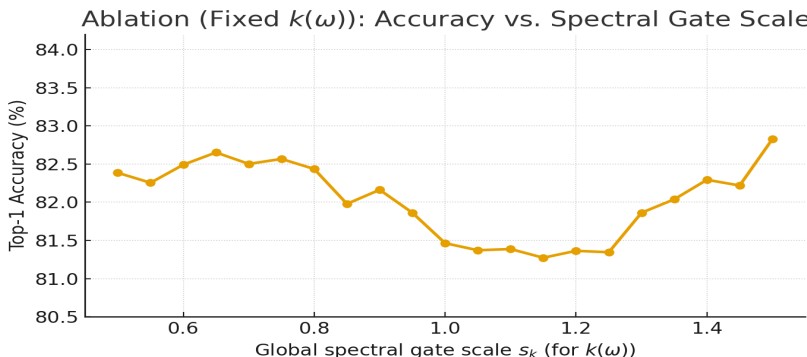

Figure 4: Ablation on spectral gate $k(\omega)$.

## A.11 SPATIAL–SPECTRAL DIFFUSION

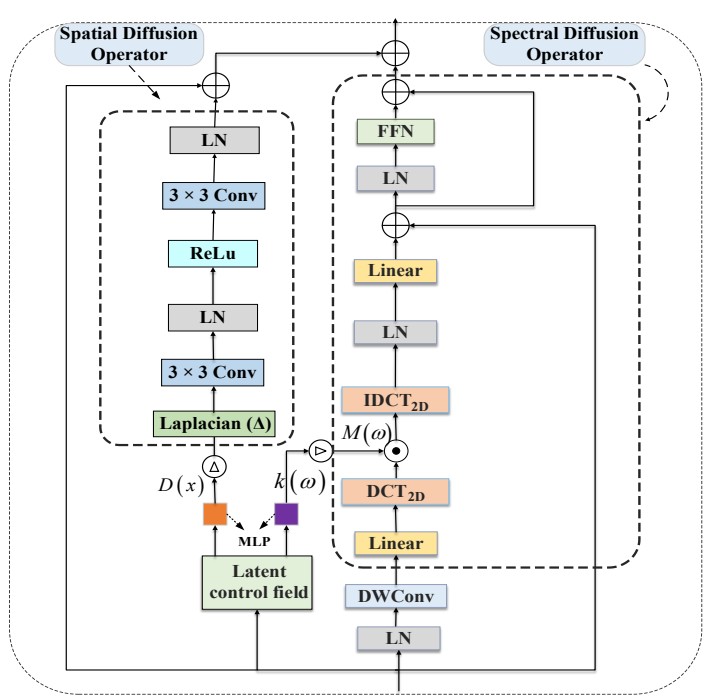

Figure 5: Overview of Spatial–Spectral Diffusion Block (S2D Block).

### A.12 AN ANALYSIS OF THE SPATIAL–SPECTRAL DIFFUSION OPERATOR

#### A.12.1 REPRESENTATION ANALYSIS OF THE SPATIAL–SPECTRAL DIFFUSION OPERATOR.

**Spatial Only.** Anisotropic diffusion suppresses cross-edge propagation and diffuses along the principal structural axes, making responses better aligned with object contours. In the cat/dog/cyclist examples, boundaries and key parts (e.g., cat ears and nose, rider arms and bike frame) become more sharply focused, though some scattered background activations remain.

**Spectral Only.** The spectral heat kernel selectively attenuates spurious high frequencies and texture noise, substantially reducing background activations and turning spotty attention into coherent blobs (notably for dog faces and small birds within foliage). Meanwhile, contour sharpness is slightly weaker than in the spatial path.

**Spatial–Spectral Diffusion (S2D).** When spatial and spectral operators are used jointly, attention concentrates on the most discriminative regions (cat eyes/nose, dog eye regions, bird torso, cyclist helmet and upper body) while maintaining a clean background and crisp boundaries—achieving a balanced trade-off between detail preservation and noise suppression. Compared with the baseline and single-path variants, S2D is more stable in complex backgrounds (e.g., foliage), small-object cases, and multi-part structures (person + bicycle), and its heatmaps align more closely with the semantic targets.

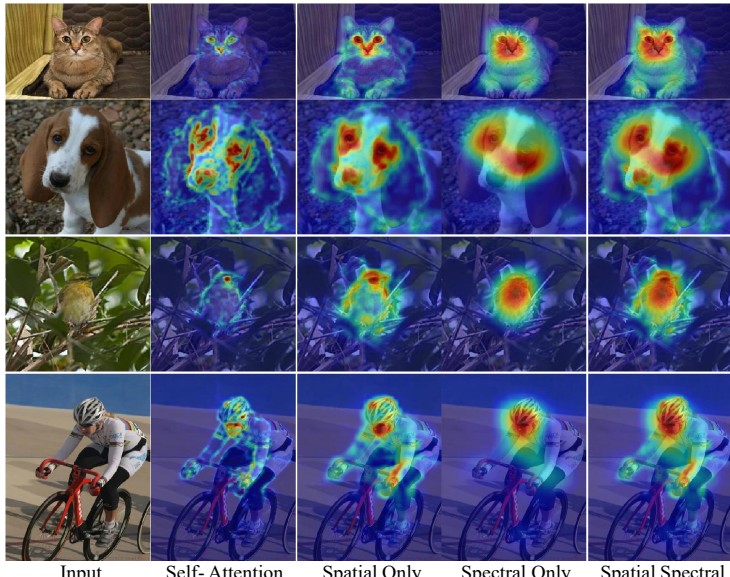

Figure 6: Visualization of the feature before/after Spatial–Spectral Diffusion Operator with ImageNet-1KRussakovsky et al. (2015) classification pre-trained Ours model.

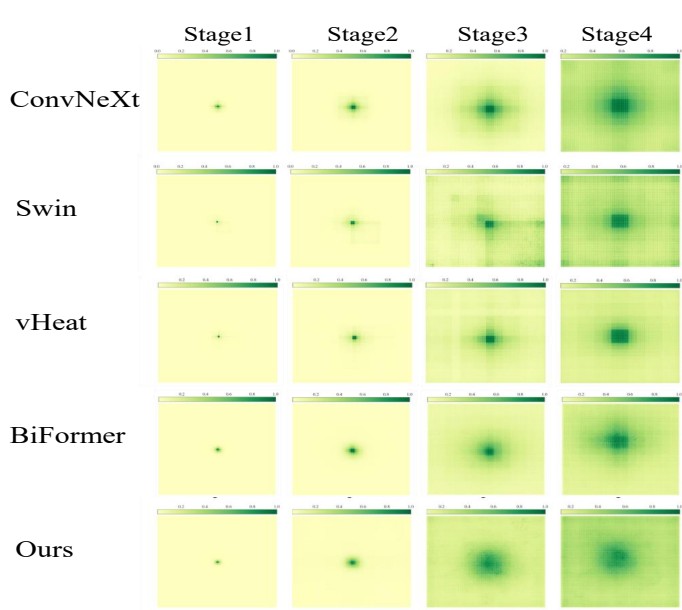

Figure 7: Visualization of the Effective Receptive Field (ERF) on ImageNet-1K Russakovsky et al. (2015). ConvNeXt-B Liu et al. (2022), Swin-B Liu et al. (2021), vHeat-BWang et al. (2025b), BiFormer-B Zhu et al. (2023), Ours.

## A.13   DOWNSTREAM TASK

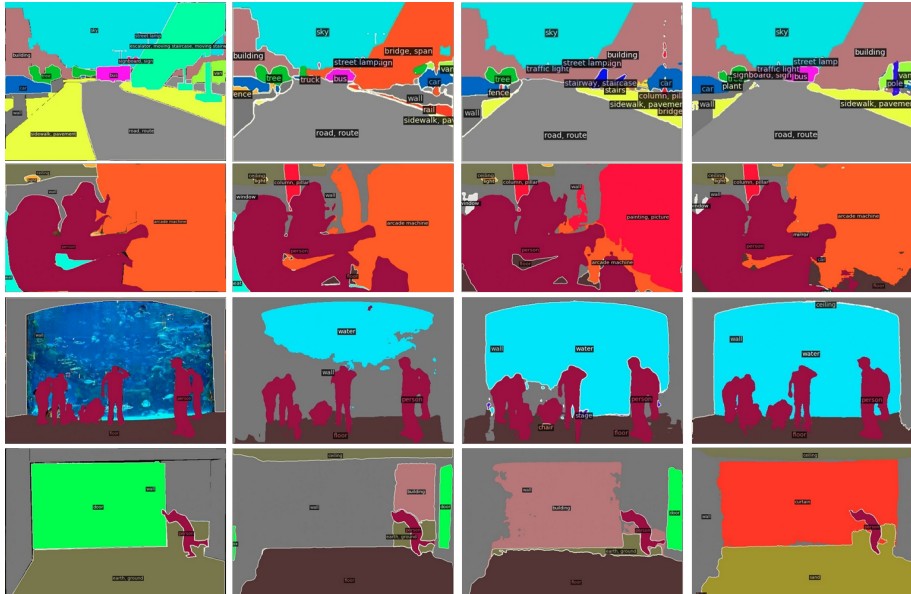

Figure 8: Qualitative results for semantic segmentation on ADE20KZhou et al. (2017).From left to right is ConvNeXt-B Liu et al. (2022), Swin-B Liu et al. (2021), Ours.

