# OpenReview forum: "UniDiff: Spectral–Spatial Vision Models under Unified Diffusion"
_ICLR.cc/2026/Conference — ICLR 2026 Conference Withdrawn Submission_

### Official Review · Reviewer_63tH · 2025-10-30

**Soundness:** 2
**Presentation:** 3
**Contribution:** 2
**Rating:** 4
**Confidence:** 4

**Summary:**

This paper introduces UniDiff, a novel vision backbone architecture designed to unify global context modeling and local boundary preservation within a single, physics-informed framework. The core of the model is the Spatial–Spectral Diffusion (S2D) block, which operates on two parallel paths derived from a unified diffusion semigroup principle:

1. A spectral path that performs a global, low-pass filtering operation in the frequency domain using a learnable heat-kernel multiplier. This is achieved efficiently via DCT/IDCT with a complexity of $O(N \log N)$, aiming to enforce global semantic consistency and suppress noise.

2. A spatial path that implements an anisotropic diffusion process with an explicit Euler step. This path is designed to preserve sharp boundaries and fine-grained details by diffusing features tangentially to edges while suppressing diffusion across them.

A key innovation is the use of a shared latent control field that predicts the parameters for both the spectral attenuation kernel $k(\omega)$ and the spatial diffusivity tensor $D(x)$. To ensure the two paths are consistent approximations of the same underlying physical process, the authors introduce three novel consistency losses: a PDE residual loss ($L_{PDE}$), an energy consistency loss ($L_{energy}$), and a boundary flux constraint ($L_{edge}$). The proposed UniDiff models demonstrate state-of-the-art or competitive performance on ImageNet-1K classification, COCO object detection/instance segmentation, and ADE20K semantic segmentation, often with improved efficiency in terms of parameters and FLOPs compared to peer models.

**Strengths:**

1. Principled and Novel Unification: The core strength of the paper is the elegant unification of global (spectral) and local (spatial) processing under a single diffusion semigroup framework. This is a departure from heuristic designs that simply stack or parallelize different types of blocks. The physics-informed constraints ($L_{PDE}$, $L_{energy}$) that enforce this unification are a key intellectual contribution.

2. Strong Theoretical Foundation: The method is exceptionally well-grounded in mathematical physics. The authors provide rigorous justifications for their design choices, with detailed proofs and derivations in the appendix. This level of rigor lends significant credibility to the work and ensures the model's properties (e.g., stability, energy dissipation) are well-understood.

3. Excellent Empirical Performance and Efficiency: The paper demonstrates SOTA or highly competitive results on ImageNet classification, COCO detection, and ADE20K segmentation. Notably, these results are often achieved with fewer parameters and FLOPs than competing methods, showcasing a superior accuracy-efficiency trade-off.

4. High-Quality Presentation and Clarity: The paper is exceptionally well-written and organized. The complex ideas are communicated with remarkable clarity, aided by excellent figures (especially Fig. 2) and a comprehensive appendix. This makes the work accessible and convincing.

5. Thorough Ablation Studies: The ablation study in Table 2 systematically validates the contribution of each component of the proposed S2D block and the associated consistency losses. This provides strong evidence that the performance gains are a direct result of the proposed design and not just a product of incidental engineering.

**Weaknesses:**

1. Complexity of the S2D Block: While the asymptotic complexity is favorable, the S2D block itself is quite intricate. It involves a DCT, an IDCT, multiple convolutions, MLPs for parameter prediction, and the assembly and application of both spectral and spatial operators. This practical complexity might represent a non-trivial constant factor in runtime and could be a barrier to widespread adoption and future development by other researchers.

2. Under-specified Latent Control Field Generation: The paper posits the "latent control field" $g(x)$ as a critical component that drives both diffusion paths. However, its generation is only briefly described in Figure 2's caption as coming from a DWConv and LN. Given its central role in encoding the geometric and semantic cues for diffusion, this mechanism feels somewhat simplistic and under-explored. A deeper analysis or ablation on the design of the control field generator would have strengthened the paper.

3. Lack of Discussion on Key Hyperparameters: The method introduces three new consistency losses, each presumably with its own weight. The paper does not discuss how these weights were determined or how sensitive the model's performance is to them. This is a crucial detail for reproducibility. Similarly, the practical handling of the CFL condition for the spatial step's time-step $\Delta t$ is not addressed, which is another important implementation detail.

4. Marginal Contribution of the Boundary Loss: According to the ablation in Table 2, the final boundary consistency loss ($L_{edge}$) provides a very small performance improvement (e.g., +0.07% Top-1 on ImageNet, +0.1 APb on COCO). While positive, this raises questions about its cost-benefit trade-off, as it requires the additional complexity of identifying boundary pixels and computing normal fluxes. The authors could comment on whether this component is truly essential.

**Questions:**

1. Could the authors please elaborate on the generation of the latent control field $g(x)$? The caption of Figure 2 mentions a DWConv, but is this single layer sufficient to capture the complex structural information needed to parameterize both the global spectral kernel and the local diffusion tensor? Have you experimented with more complex generators for $g(x)$?

2. The three consistency losses ($L_{PDE}$, $L_{energy}$, $L_{edge}$) are a core part of the method. How were their respective loss weights chosen? Could you provide some insight into the model's sensitivity to these hyperparameters?

3. Regarding the spatial diffusion step, the time step $\Delta t$ must satisfy a CFL condition that depends on the maximum value of the learned diffusivity $\kappa_{\parallel}(x)$. How is this handled in practice during training with batches of data, where $\max(\kappa_{\parallel}(x))$ could vary for each image in the batch? Is a single, conservative $\Delta t$ used for all samples, or is there an adaptive mechanism?

4. The ablation study shows that the boundary consistency loss ($L_{edge}$) offers a marginal performance gain. Could you discuss the trade-off between this small gain and the added complexity of implementing this loss (e.g., detecting boundary pixels)? In your opinion, is this component critical to the model's success?

---

### Official Review · Reviewer_4Avc · 2025-10-30

**Soundness:** 1
**Presentation:** 1
**Contribution:** 1
**Rating:** 2
**Confidence:** 4

**Summary:**

This paper introduces a physics-inspired vision backbone that incorporates a parallel Spatial Spectral Diffusion module. Concretely, the spectral branch applies a heat-kernel multiplier to achieve global low-pass homogenization, while the spatial branch performs anisotropic diffusion to preserve boundaries, further constrained by a PDE residual and energy-consistency term. Based on the reported results, the method demonstrates advantages in parameter count, FLOPs, and inference throughput compared with peer baselines.

**Strengths:**

1. The physical inspired method for vision backbone design is interesting.

**Weaknesses:**

1. The paper is difficult to follow and contains many unclear and obscure expressions. For example, what is the definition or explanation of `latent control field`, `physical clock`, `smear-sharpen counteraction`, `time-scale drift` in the abstract?

2. The authors claim that one advantage of the method is its O(N log N) computational complexity, but there is no substantive comparison to related work, including [1].

3. Key experimental details and qualitative evidence are missing, which makes the results unconvincing. For example, the abstract claims robustness for detection and segmentation tasks, yet no detection visualization is provided and the segmentation visualizations shown in Fig. 8 are weak. Hyperparameters for training are not reported, nor is the full loss formulation with coefficients for each component provided.

4. In Lines 693-695, the authors state that "the division is ... numerically stable and practically efficient" but no experimental evidence is given to support this claim.

[1] Log-Linear Attention. Arxiv, 2025.

**Questions:**

see weakness

---

### Official Review · Reviewer_ubRa · 2025-10-31

**Soundness:** 3
**Presentation:** 2
**Contribution:** 3
**Rating:** 4
**Confidence:** 4

**Summary:**

This paper introduces UniDiff, a physics-inspired vision backbone that unifies spectral and spatial diffusion within one framework. It models feature propagation as a diffusion process, where the spectral path performs global smoothing via a heat-kernel filter and the spatial path applies anisotropic diffusion to preserve boundaries. A shared latent control field jointly governs both paths under physical constraints of PDE residual, energy consistency, and boundary flux. Experimental results on ImageNet classification, ADE20K semantic segmentation, and COCO object detection validate that UniDiff achieves higher accuracy and efficiency than existing vision backbones, including ConvNeXt, Swin, and vHeat.

**Strengths:**

1. Compared to vHeat, UniDiff introduces a spatial diffusion operator in addition to the spectral path, effectively enhancing instance boundary perception, as evidenced by clearer edge details in Fig. 6.
2. The model enforces three physical consistency losses—PDE residual, energy consistency, and boundary flux—which align the spectral and spatial paths under a shared diffusion process. These constraints respectively ensure update coherence, stable energy dissipation, and boundary preservation, forming a physically grounded coupling mechanism. Ablation results show progressive performance gains when these losses are added, confirming their effectiveness in improving both stability and visual fidelity.

**Weaknesses:**

1. The paper suffers from several presentation issues, such as citations appearing in the abstract and mislabeled entries in Tables 5–7, where TransNeXt is incorrectly marked as Ours, raising concerns about proofreading and result reliability.
2. The S2D Block illustration (Fig. 2) lacks critical details—specifically, the origin and computation of the latent control field and the meaning of the orange and purple blocks—making the module-level design unclear.
3. Due to the above presentation issues and the lack of a Reproducibility Statement in both the main paper and supplementary materials, the credibility and reproducibility of the reported experimental results are questionable.
4. Although the paper claims to achieve robust visual representations, no experiments are conducted on robustness-oriented benchmarks such as ImageNet-A/R/S/C or ObjectNet, leaving the robustness claim unsubstantiated.

**Questions:**

Please refer to the weaknesses.

---

### Official Review · Reviewer_kic4 · 2025-10-31

**Soundness:** 3
**Presentation:** 3
**Contribution:** 3
**Rating:** 4
**Confidence:** 4

**Summary:**

This paper proposes a novel general-purpose vision backbone named UniDiff. Its core is a unified "Spatial-Spectral Diffusion" (S2D) module that operates within a diffusion semigroup framework. The S2D module innovatively parallels two paths: a spectral path, which utilizes a heat-kernel multiplier for efficient ($O(N \log N)$ complexity) global low-pass filtering and semantic homogenization; and a spatial path, which uses anisotropic diffusion to preserve local boundaries and fine structures. These two paths are governed by parameters predicted from a shared latent control field and are jointly regulated by physics-informed constraints, such as PDE residual and energy consistency, to ensure they collaborate under the same physical clock and avoid the common "smear-sharpen" counteraction.

**Strengths:**

- The primary contribution of this paper is a novel and physics-inspired UniDiff framework. Unifying global semantic consistency (via an $O(N \log N)$ complexity spectral heat-kernel) and local boundary fidelity (via spatial anisotropic diffusion) under a single diffusion semigroup is a highly insightful idea, offering a theoretically coherent and efficient solution to a long-standing challenge in vision modeling.
- The experimental results are highly sufficient and competitive. UniDiff demonstrates SOTA or near-SOTA performance on ImageNet classification as well as on downstream dense prediction tasks like COCO and ADE20K, all while showing advantages in parameter count and computational cost.

**Weaknesses:**

- Ambiguous Description of the "Latent Control Field": The $g(x)$ is a core component driving both paths, but it is insufficiently described in the main text. It is described as simultaneously predicting global spectral parameters $k(w)$ (via global pooling and an MLP) and local spatial tensors $D(x)$ (pixel-wise). However, how $g(x)$ itself is computed from the input features is not clearly explained in Figure 2 or Sections 3.4/3.5. This makes the model's key mechanism difficult to fully understand and reproduce.
- Questionable Rationale for the PDE Residual Loss ($L_PDE$): This is a key constraint in the paper. Equation 14 attempts to align the time derivative of the spectral update (a global smoothing operation), $ (u\_tilde^{l+1} - u^l) / t_g $, with the spatial generator $Lu^l$ (a local anisotropic operation). The justification for this approximation, $ (u\_bar^{l+1} - u^l) / t_g \approx Lu^l $, which is based on a first-order Taylor expansion, lacks sufficient theoretical support in the main text, especially when $D(x)$ varies sharply in space, where the approximation error could be large. The precise definition of $L\_tilde$ is also unclear.
- Missing Implementation Details for the Boundary Flux Constraint ($L_edge$): This loss (Equation 16) relies on a summation over a boundary pixel set $E$. The authors state that $E$ is "obtained, e.g., by a structure-tensor or gradient-magnitude threshold." This introduces a non-trivial external module or hyperparameter. How sensitive is the model's performance (especially for segmentation tasks) to the specific choice of this boundary detector and its parameters (e.g., the threshold)? The ablation study (Table 2) only ablates the $L_edge$ loss itself, not the impact of its implementation.
- Insufficient Comparison with Related Work: The paper mentions other heat-equation-inspired works like vHeat. vHeat appears to correspond only to UniDiff's "spectral path." The paper needs to more clearly articulate the fundamental advantages of introducing the anisotropic spatial path and the specific physical coupling losses compared to vHeat. Furthermore, $O(N \log N)$ complexity also exists in other methods (e.g., FFT-based global convolution, Mamba). The paper lacks a discussion on why this diffusion-based approach is superior for vision tasks compared to these other efficient global modeling paradigms.

**Questions:**

1. Regarding the implementation of $g(x)$: Could the authors please detail how the "latent control field" $g(x)$ is computed from the S2D block's input features? What is its specific network architecture (e.g., a simple 1x1 convolution, or a more complex MLP)?
2. Regarding the definition of $L_PDE$: In the $L_PDE$ (Equation 14), what is the precise mathematical definition of $L\_tilde \cdot u^l$? Is it simply the spatial diffusion operator div(D(x) * grad(u^l))? If so, why is it reasonable to force the result of a global low-pass filter (spectral path) to align with that of a local anisotropic operator (spatial path)?
3. Regarding the sensitivity of $L_edge$: In the experiments, how was the boundary set $E$ for $L_edge$ specifically implemented? For instance, if a gradient magnitude threshold was used, how was this threshold set? Is the model sensitive to the choice of this threshold?
4. Regarding the CFL Stability Condition: The explicit Euler step in the spatial path is constrained by the CFL condition (Equations 11 and 40), which depends on the maximum value of $k\_parallel(x)$. How was $Delta\_t$ set during training? Was it fixed, or dynamically adjusted based on the predicted $D(x)$? If the model learns a very large $k\_parallel(x)$, could this cause $Delta\_t$ to approach zero, thereby nullifying the contribution of the spatial path?
5. Regarding the Ablation Study: The ablation in Table 2 is helpful, but a key comparison is missing. Could the authors provide the performance of a "Spatial Only" model (Baseline + Spatial Operator + Losses)? This would allow for a fair comparison against the "Spectral Only" model (Baseline + Spectral Operator) in row 2 of Table 2, thus more clearly disentangling the respective contributions of the two paths.

---

### Note · Authors · 2025-11-16

I have read and agree with the venue's withdrawal policy on behalf of myself and my co-authors.